# Maturation of Purkinje cell firing properties relies on neurogenesis of excitatory neurons

**Meike E van der Heijden[1,2], Elizabeth P Lackey[1,2,3], Ross Perez[2], Fatma S Işleyen[1,4], Amanda M Brown[1,2,3], Sarah G Donofrio[1,2,3], Tao Lin[1,2], Huda Y Zoghbi[2,3,4,5], Roy V Sillitoe[1,2,3,4,6]\***

[1]Department of Pathology and Immunology, Baylor College of Medicine, Houston, United States; [2]Jan and Dan Duncan Neurological Research Institute at Texas Children's Hospital, Houston, United States; [3]Department of Neuroscience, Baylor College of Medicine, Houston, United States; [4]Program in Developmental Biology, Baylor College of Medicine, Houston, United States; [5]Howard Hughes Medical Institute, Department of Molecular and Human Genetics, Baylor College of Medicine, Houston, United States; [6]Development, Disease Models and Therapeutics Graduate Program, Baylor College of Medicine, Houston, United States

**Abstract** Preterm infants that suffer cerebellar insults often develop motor disorders and cognitive difficulty. Excitatory granule cells, the most numerous neuron type in the brain, are especially vulnerable and likely instigate disease by impairing the function of their targets, the Purkinje cells. Here, we use regional genetic manipulations and in vivo electrophysiology to test whether excitatory neurons establish the firing properties of Purkinje cells during postnatal mouse development. We generated mutant mice that lack the majority of excitatory cerebellar neurons and tracked the structural and functional consequences on Purkinje cells. We reveal that Purkinje cells fail to acquire their typical morphology and connectivity, and that the concomitant transformation of Purkinje cell firing activity does not occur either. We also show that our mutant pups have impaired motor behaviors and vocal skills. These data argue that excitatory cerebellar neurons define the maturation time-window for postnatal Purkinje cell functions and refine cerebellar-dependent behaviors.

**\*For correspondence:**
sillitoe@bcm.edu

## Introduction

Abnormal cerebellar development instigates motor diseases and neurodevelopmental disorders including ataxia, dystonia, tremor, and autism. These conditions have a high incidence in premature infants as well as in newborns with cerebellar hemorrhage (*Dijkshoorn et al., 2020*; *Limperopoulos et al., 2007*; *Steggerda et al., 2009*; *Zayek et al., 2012*). The affected children ultimately attain a smaller cerebellar size compared to children born full-term (*Limperopoulos et al., 2005*; *Volpe, 2009*), and they have altered functional connectivity between the cerebellum and the neocortex (*Herzmann et al., 2019*; *Hortensius et al., 2018*). Observations from human clinical data indicate a strong correlation between abnormalities in cerebellar size and the subsequent incidence of cognitive disorders, suggesting that the late-term stage of cerebellar development encompasses a critical developmental time-window for establishing not only the motor but also the non-motor circuitry that is required for complex cerebellar functions and behavior (*van der Heijden et al., 2021b*; *Volpe, 2009*; *Wang et al., 2014*).

The cerebellum undergoes a series of dynamic developmental changes during the third trimester and early postnatal stages, a period that corresponds to the first 2 postnatal weeks in mice

**eLife digest** Preterm infants have a higher risk of developing movement difficulties and neurodevelopmental conditions like autism spectrum disorder. This is likely caused by injuries to a part of the brain called the cerebellum. The cerebellum is important for movement, language and social interactions. During the final weeks of pregnancy, the cerebellum grows larger and develops a complex pattern of folds. Tiny granule cells, which are particularly vulnerable to harm, drive this development.

Exactly how damage to granule cells causes movement difficulties and other conditions is unclear. One potential explanation may be that granule cells are important for the development of Purkinje cells in the brain. The Purkinje cells send and receive messages and are very important for coordinating movement.

To learn more, van der Heijden et al. studied Purkinje cells in mice during a period that corresponds with the third trimester of pregnancy in humans. During this time, the pattern of electrical signals sent by the Purkinje cells changed from slow and irregular to fast and rhythmic with long pauses between bursts. However, mice that had been genetically engineered to lack most of their granule cells showed a completely different pattern of Purkinje cell development. The pattern of electrical signals emitted by these Purkinje cells stayed slow and irregular. Mice that lacked granule cells also had movement difficulties, tremors, and abnormal vocalizations.

The experiments confirm that granule cells are essential for normal brain development. Without enough granule cells, the Purkinje cells become stuck in an immature state. This discovery may help physicians identify preterm infants with motor disorders and other conditions earlier. It may also lead to changes in the care of preterm infants designed to protect their granule cells.

(*Sathyanesan et al., 2019*). At the level of gross morphology, the rapid proliferation of granule cell precursors and the integration of granule cells into the cerebellar circuit result in a fivefold enlargement of cerebellar size (*Chang et al., 2000*), an increase in foliation complexity (*Corrales et al., 2006*), and the formation of the tri-laminar cerebellar cortex (*Miyata et al., 2010*). At the cellular level, the differentiation and arrival of mature granule cells into the circuit modifies the laminar localization and synaptic specification of cerebellar afferents that project to Purkinje cells (*van der Heijden and Sillitoe, 2021*). Despite an extensive literature describing the temporal anatomical changes in the cerebellum and studies postulating that perturbations to the cerebellum during this dynamic period may lead to cognitive, social, and motor deficits (*Volpe, 2009*; *Wang et al., 2014*), there is still relatively little known about how the anatomical aspects of cerebellar development correspond to its functional maturation programs that establish animal behavior.

Previous studies show that preterm birth in pigs, as well as postnatal hemorrhage and hypoxia (which mimics cerebellar hemorrhage) in mice, result in decreased granule cell numbers (*Iskusnykh et al., 2018*; *Yoo et al., 2014*) and abnormal motor control (*Sathyanesan et al., 2018*; *Yoo et al., 2014*). Importantly, postnatal hypoxia causes impairments in the spontaneous firing properties of Purkinje cells in mice (*Sathyanesan et al., 2018*), and impaired Purkinje cell firing properties are also observed in prematurely born baboons (*Barron and Kim, 2020*). It is not surprising that decreasing the number of granule cells results in abnormal Purkinje cell function, since each Purkinje cell may integrate inputs from up to two hundred fifty thousand excitatory granule cell synapses (*Huang et al., 2014*; *Napper and Harvey, 1988*). Nevertheless, genetic silencing of granule cells causes only modest alterations to the baseline firing properties of Purkinje cells (*Galliano et al., 2013*), perhaps because the predominant Purkinje cell action potential, called the simple spike, is spontaneously generated by the Purkinje cells (*Raman and Bean, 1999*). In line with this hypothesis, gross motor control is not affected in mice with impaired granule cell signaling, although they do have impaired motor learning (*Galliano et al., 2013*). In contrast, some mouse models that lack the majority of granule cells (agranular mice) have such severe motor defects that these mice are named after their phenotype; for example *scrambler*, *weaver*, *reeler*, and *staggerer* (*Bolivar et al., 1996*; *Falconer, 1951*; *Heuzé et al., 1997*; *Le Marec et al., 1997*; *Sheldon et al., 1997*). The discordance between the functional outcomes in mutant mice with reduced granule cell numbers and mice

lacking granule cell signaling raises the question whether granule cell neurogenesis, rather than granule cell synaptic signaling, contributes more to the maturation of Purkinje cell firing properties in vivo.

To probe these cellular interactions, we manipulated the mouse cerebellum by genetically blocking neurogenesis of excitatory neurons. We used an *En1*<sup>Cre</sup> allele (*Wurst et al., 1994*) to delete the proneural gene, *Atoh1*, from the embryonic hindbrain. *Atoh1* is necessary for the development of excitatory cerebellar neurons (*Ben-Arie et al., 1997*; *Rose et al., 2009*) and is not expressed in the inhibitory Purkinje cells. In this agranular cerebellar model, we test how Purkinje cells develop their structure, connectivity, and function using immunostaining, neural tract tracing, and in vivo electrophysiology performed in the second postnatal week. We also examined motor function and vocal skills in the mutant pups to test how the structural and functional changes in the cerebellum impact the expression of behaviors that are usually acquired during the first two weeks of life.

## Results

### *Atoh1* lineage neurons shape multiple aspects of cerebellar morphogenesis

To test the hypothesis that granule cell neurogenesis is essential for the functional development of Purkinje cells, we first needed to establish an appropriate agranular mouse model. In previously generated models, the mice lack granule cells due to spontaneously occurring mutations in genes with widespread expression patterns (*Falconer, 1951*; *van der Heijden and Sillitoe, 2021*; *Sheldon et al., 1997*). Therefore, one could not differentiate cell-extrinsic from cell-intrinsic effects in Purkinje cells (*Dusart et al., 2006*; *Gold et al., 2007*). To impair granule cell neurogenesis in a manner independent of genes expressed in Purkinje cells, we made use of the distinct origins of Purkinje cells and granule cells to establish a new model of agranular mice (*Figure 1A–B*; *Hoshino et al., 2005*; *Rose et al., 2009*). *Atoh1* is necessary for the development of excitatory cerebellar neurons, including granule cells (*Ben-Arie et al., 1997*), which are the most populous cell type in the cerebellum and make up at least 99% of all neurons in the cerebellum (*Consalez et al., 2020*). *En1* is a homeodomain transcription factor that is expressed in the mesencephalon and rhombomere 1 by embryonic day 8 (E8), where it is required for the formation of the cerebellum (*Davis and Joyner, 1988*; *Wurst et al., 1994*). The intersection of the *Atoh1* and *En1* lineages converges on excitatory neurons that originate from the rostral rhombic lip (rRL) (*Wang et al., 2005*), but does not include inhibitory neurons like Purkinje cells that originate in the ventricular zone (*Hoshino et al., 2005*). We demonstrate this selectivity by conditional expression of TdTomato, which is expressed by an intersectional reporter allele that is dependent on both Cre- (*En1* [*Wurst et al., 1994*]) and FlpO- (*Atoh1* [*van der Heijden and Zoghbi, 2018*]) mediated excision of a stop-cassette (*Figure 1C*). As expected, we observed TdTomato expression in the granule cell layer and molecular layer of the cerebellar cortex, where the axons (parallel fibers) of granule cells reside, but not in the Purkinje cell layer (*Figure 1C*).

Our rationale for choosing to impair granule cell neurogenesis by employing a conditional deletion of *Atoh1* from the *En1* domain was because *Atoh1* null mice are neonatal lethal (*Ben-Arie et al., 1997*). We generated conditional knockout mice by crossing mice that are heterozygous for the *En1*<sup>Cre</sup> knock-in allele (*Wurst et al., 1994*) and heterozygous for *Atoh1* with mice that are homozygous for a LoxP-flanked *Atoh1* allele (*Atoh1*<sup>fl/fl</sup>) (*Shroyer et al., 2007*). From this cross, we obtain control mice with one of the following three genotypes: *Atoh1*<sup>fl/+</sup> mice, *Atoh1*<sup>FlpO/fl</sup> mice, *En1*<sup>Cre/+</sup>; *Atoh1*<sup>fl/+</sup> mice. The different control alleles were analyzed separately for the behavioral data and subsequently grouped together for anatomical and electrophysiological analyses due to the absence of any significant behavioral differences. In addition, the cross generates our experimental *En1*<sup>Cre/+</sup>; *Atoh1*<sup>fl/FlpO</sup> mice that we will hereafter refer to as *En1*<sup>Cre/+</sup>;*Atoh1*<sup>fl/-</sup> mice because the *Atoh1*<sup>FlpO</sup> is a functional null-allele (*van der Heijden and Zoghbi, 2018*). We previously showed that these *En1*<sup>Cre/+</sup>;*Atoh1*<sup>fl/-</sup> mice are informative for assessing the period of dynamic cerebellar development (P7-P14) but the animals succumb at around weaning age (P21), likely due to a combination of impaired motor coordination and breathing abnormalities (*van der Heijden and Zoghbi, 2018*). Using these mice, we have now investigated how the *En1*-mediated deletion of *Atoh1* influences the period of cerebellar growth that usually occurs in the second postnatal week. We found that

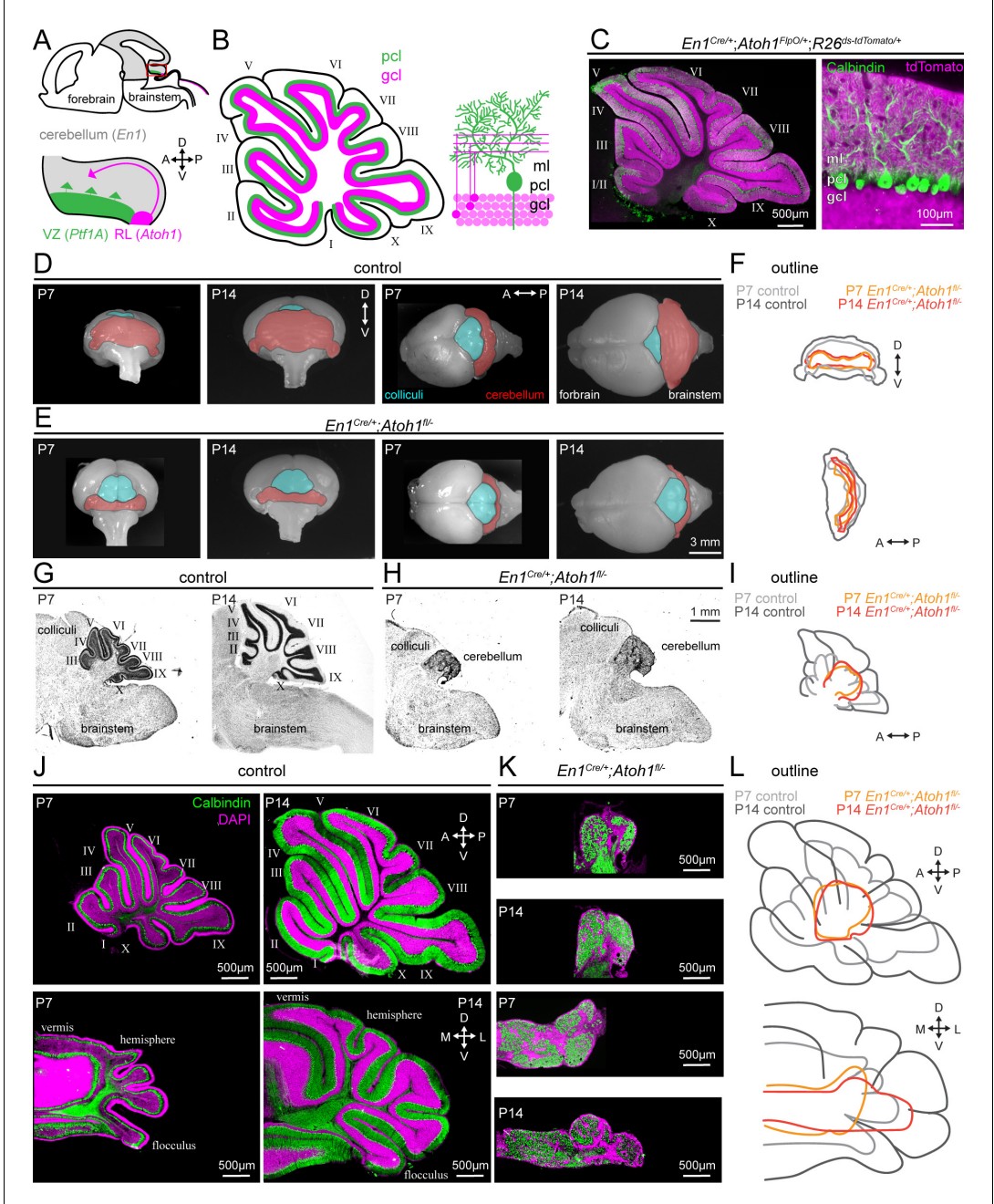

**Figure 1.** Conditional deletion of *Atoh1* from the *En1* domain prevents proper size expansion, lamination, and foliation of the cerebellum. (**A**) Schematic of an embryonic brain. Inset is the cerebellar anlage. *Atoh1* domain (excitatory lineage neurons, including granule cell precursors, pink), *Ptf1a* domain (inhibitory lineage neurons, including Purkinje cell precursors, green), *En1* domain (gray). Orientation is the same for all panels unless otherwise indicated. (**B**) Schematic of a sagittal section of a P14 cerebellum (left). Cerebellar lobules are labeled with Roman numerals (*Larsell, 1952*). Schematic of the cerebellar microcircuit with Purkinje cells and granule cells (right). Purkinje cell = green; granule cell = pink; ml = molecular layer, pcl = Purkinje cell layer; gcl = granule cell layer. (**C**) Intersectional labeling of *En1;Atoh1* domain with tdTomato (pink) shows no overlap with Purkinje cells (Calbindin; green). (**D**) Whole brain images of cerebellum from anterior (left two images) and dorsal view (right two images) from P7 and P14 in control animals. The cerebellum is pseudo-colored in red. (**E**) Whole brain images of the cerebellum from anterior (left two images) and dorsal views (right two images) from P7 and P14 *En1^{Cre/+}*;*Atoh1^{fl/-}* mice. The cerebellum is pseudo-colored in red. (**F**) Outline of cerebellar size in control and *En1^{Cre/+}*;*Atoh1^{fl/-}* mice at P7 and P14 showing size expansion in control, but not *En1^{Cre/+}*;*Atoh1^{fl/-}* mice. (**G**) Sagittal sections of the hindbrain from control mice stained with cresyl violet to visualize gross architecture and regional organization of the brain. (**H**) Sagittal sections of the hindbrain from *En1^{Cre/+}*;*Atoh1^{fl/-}* mice stained with cresyl violet showing the general deformities in regional patterning. (**I**) Outline of cerebellar size in control and *En1^{Cre/+}*;*Atoh1^{fl/-}* mice at P7 and P14 showing size expansion in control, but not *En1^{Cre/+}*;*Atoh1^{fl/-}* mice. (**J**) Sagittal (top) and coronal (bottom) whole section images of cerebella from P7

*Figure 1 continued on next page*

*Figure 1 continued*

(left) and P14 (right) control mice. Purkinje cells are stained with Calbindin (green) and DAPI (pink). Note the layered and foliated structure of the cerebellum. (K) Sagittal (top) and coronal (bottom) whole section images of cerebella from P7 and P14 *En1^{Cre/+};Atoh1^{fl/-}* mice. Purkinje cells are stained with Calbindin (green) and DAPI (pink). All images are presented at the same magnification. Note the lack of layers and foliation. All images in J and K are presented at the same magnification. (L) Outline of cerebellar size in control and *En1^{Cre/+};Atoh1^{fl/-}* mice at P7 and P14 showing size expansion in control, but not in the *En1^{Cre/+};Atoh1^{fl/-}* mice. All images are representative of N=3 per age and genotype group.

The online version of this article includes the following source data and figure supplement(s) for figure 1:

**Figure supplement 1.** Conditional deletion of *Atoh1* from the *En1* domain leads to cerebellum-specific morphological abnormalities.

**Figure supplement 2.** Conditional deletion of *Atoh1* from the *En1* domain reduces the density of excitatory cerebellar cell types, but increases the density of inhibitory cerebellar cell types.

**Figure supplement 3.** Conditional deletion of *Atoh1* from the *En1* domain reduces the density of *En1;Atoh1* lineage neurons.

**Figure supplement 4.** Conditional deletion of *Atoh1* from the *En1* domain reduces the number of unipolar brush cells and excitatory nuclei neurons.

**Figure supplement 4—source data 1.** Source data and specific p-values for representative graphs in *Figure 1—figure supplement 4*.

cerebellar size rapidly expands between P7 and P14 in control mice (*Figure 1D*, cerebellum is pseudo-colored in red) but that no such size-expansion occurs in the *En1^{Cre/+};Atoh1^{fl/-}* mice (*Figure 1E*). When we overlay the outlines of the cerebellum, we furthermore observe that the size of the cerebellum in *En1^{Cre/+};Atoh1^{fl/-}* mice is markedly smaller at both P7 and P14 compared to that of similar aged controls (*Figure 1F*). In sagittal tissue sections cut through the hindbrain and then stained with cresyl violet to visualize cell nuclei, we also observe that P7 and P14 *En1^{Cre/+};Atoh1^{fl/-}* cerebella lack a densely packed granule cell layer, that *En1^{Cre/+};Atoh1^{fl/-}* cerebella do not form cerebellar lobules, and that there is no substantial increase in cerebellar size that takes place during the second postnatal week, which are morphological milestones typically observed in control mice (*Figure 1G–I*). At P14, sagittal cerebellar tissue sections from *En1^{Cre/+};Atoh1^{fl/-}* mice measure just 11.3% (±1%, n=12 sets) the size of paired control tissue sections and in addition, coronal-cut cerebellar tissue sections from *En1^{Cre/+};Atoh1^{fl/-}* mice measure just 20.2% (±3%, n=6 sets) the size of paired control tissue sections. Moreover, as expected based on the expression patterns of the two genes we used in our manipulation, the deletion of *Atoh1* from just the *En1* domain did not result in any gross morphological abnormalities in brain regions other than the cerebellum, including the olfactory bulb, hippocampus, striatum, thalamus, pons, vestibular nuclei, inferior olive, and neocortex (*Figure 1—figure supplement 1*).

Previous studies on agranular animal models have reported a spectrum of morphological changes in the cerebellum, depending on the degree of granule cell loss; a range of cerebellar size differences compared to controls, thinner to completely absent granule cell layers, and fewer or complete lack of cerebellar foliation (compared to controls, the cerebellum in X-irradiated rats has: smaller size, thinner granule cell layer [*Armstrong and Hawkes, 2001*; *Zanjani et al., 1992*]; *staggerer* and *weaver* mice: smaller cerebellar size, thinner granule cell layer, less foliation [*Armstrong and Hawkes, 2001*; *Zanjani et al., 1992*]; *reeler* and *scrambler* mice: smaller cerebellar size, thinner granule cell layer, no foliation [*Goldowitz et al., 1997*]; *Atoh1* chimera mice: smaller cerebellar size, thinner/no granule cell layer, less/no foliation [the phenotype is variable depending on the percentage of chimerism; *Jensen et al., 2004*]). Our *En1^{Cre/+};Atoh1^{fl/-}* mice encompass all these malformations, without the variability that was observed in *Atoh1* chimera mice (*Jensen et al., 2004*) or the gross abnormalities observed outside of the cerebellum that was reported for the *reeler* and *scrambler* mice (*Goldowitz et al., 1997*). Therefore, our *En1^{Cre/+};Atoh1^{fl/-}* mice encompass all of the morphological changes observed in previous agranular models, although at the more extreme end of the severity spectrum, with little experimental variability and high regional specificity.

## *Atoh1* deletion reduces the number of excitatory lineage neurons in the cerebellum

Next, we used immunohistochemistry to investigate which of the main cellular components of the cerebellum is affected in *En1^{Cre/+};Atoh1^{fl/-}* mice. Based on the intersectional strategy we employed in our studies, we would expect a reduction in excitatory lineage cells with minimal changes to inhibitory neurons, including Purkinje cells. We show that deletion of *Atoh1* prevents granule cells from reaching differentiation by staining for GABARα6, which marks mature granule cells (*Figure 1—figure supplement 1–2A*). However, it remained unclear whether this change in signal was due to

abnormal maturation or impaired granule cell neurogenesis. To address this question, we decided to employ a genetic strategy to label all *En1;Atoh1* lineage neurons in control and conditional knockout mice with an intersectional TdTomato-reporter allele (*Figure 1—figure supplement 1–3A–B*). In this strategy, control animals are $En1^{Cre/+};Atoh1^{FlpO/+};Rosa^{lsl-lfl-TdTomato/+}$ mice (as in *Figure 1C*) and conditional knockout mice are $En1^{Cre/+};Atoh1^{FlpO/fl};Rosa^{lsl-lfl-TdTomato/+}$ mice. Within the granule cell layer of control mice, the TdTomato$^+$ signal saturates due to the dense packing of granule cell bodies. This leads to a signal density of 99.6% (± 0.1%, N=2, n=10) within the granule cell layer of lobule IX in control mice ($En1^{Cre/+};Atoh1^{FlpO/+};Rosa^{lsl-lfl-TdTomato/+}$), compared to just 43.4% (± 3.5%, N=3, n=15) in the ventral-caudal regions of cerebella in conditional knockout mice ($En1^{Cre/+};Atoh1^{FlpO/fl};Rosa^{lsl-lfl-TdTomato/+}$) (*Figure 1—figure supplement 1–3D*). Despite the mainly qualitative utility of these data, this reduced signal density does not provide direct numerical insight in the quantity of *En1;Atoh1* lineage neurons that escape our genetic manipulation because it does not account for the origins of the signal saturation, the nearly 50-fold size difference between control and conditional knockout cerebella, or the difference in cellular compartments present in the area of interest (only cell bodies and dendrites in control mice, versus cell bodies, dendrites, and axons in conditional knockout mice). Unfortunately, the signal density in the control animals prevented us from manually counting individual *En1;Atoh1* lineage neurons in control mice, but we were able to count the total number of TdTomato$^+$, *En1;Atoh1* lineage neurons in sagittal tissue sections from conditional knockout mice. We found an average of 925 (± 22, N=3, n=12) TdTomato$^+$ cells in each tissue section (*Figure 1—figure supplement 1–3A–BC–D*). We usually obtain ~70 sagittal sections from a single conditional knockout mouse brain (N=5), leading us to estimate that the total number of *En1;Atoh1* lineage cells that escape the conditional knockout of *Atoh1* from the *En1* lineage to be in the range of 65 • $10^3$ (/thousand) neurons per mutant mouse brain. The total number of cerebellar neurons (99% of which are granule cells) in control mice has previously been estimated in the range of millions (between 35 • $10^6$ and 70 • $10^6$ neurons [*Consalez et al., 2020*; *Herculano-Houzel et al., 2006*; *Surchev et al., 2007*; *Vogel et al., 1989*]). We therefore estimate an approximate $10^3$ (thousand)-fold reduction in the total number of *En1;Atoh1* lineage neurons in our mutant mice, or less than 1% of the *En1;Atoh1* lineage neurons escape the genetic manipulation.

Consistent with the substantially reduced number of *En1;Atoh1* lineage neurons in the conditional knockout mice, we observe that $En1^{Cre/+};Atoh1^{fl/-}$ conditional knockout mice have a reduced pool of excitatory unipolar brush cells, which also derive from the *En1;Atoh1* lineage (*Figure 1—figure supplement 1–4A*), compared to the density of unipolar brush cells that are localized to lobule IX and X in control mice (*Figure 1—figure supplement 2B–D*). When we counted the number Tbr2$^+$unipolar brush cells in control and conditional knockout mice, we found an approximately eight-fold reduction in the number of unipolar brush cells (*Figure 1—figure supplement 4C–E*). Interestingly, none of the Tbr2$^+$ escaper unipolar brush cells in $En1^{Cre/+};Atoh1^{FlpO/fl};Rosa^{lsl-lfl-TdTomato/+}$ conditional knockout mice express the TdTomato reporter (*Figure 1—figure supplement 4A*), suggesting that these cells may derive from a different sub-lineage. We also counted the number of excitatory cerebellar nuclei neurons, that derive from the *En1;Atoh1* lineage (*Figure 1—figure supplement 4B*) and found an approximate fivefold reduction in their number (*Figure 1—figure supplement 4F–H*). Unlike the escaper Tbr2$^+$ unipolar brush cells, cerebellar nuclei neurons in the conditional knockout mice express the TdTomato reporter, but the NFH signal appears more fibrous. Taken together, we find a significant reduction in the total number of all types of excitatory neurons that are derived from the *En1;Atoh1* lineage. We also observed an abnormal localization of GABARα6 expression in granule cells (*Figure 1—figure supplement 2A*) and NFH expression in excitatory nuclei neurons (*Figure 1—figure supplement 4F–G*), which is an interesting molecular response to progressing this far into development without *Atoh1*.

To confirm that our genetic strategy did not prevent the development of inhibitory neurons, specifically Purkinje cells, we stained control cerebella using a combination of Calbindin, which marks Purkinje cells, and DAPI, which binds to DNA and ubiquitously marks cell nuclei. In sagittal and coronal tissue sections cut through the cerebella from control mice, the staining again highlights the increase in size between P7 and P14, as well as the formation of perfectly delineated dorsal-ventral layering of cells and the establishment of the characteristic folding that defines the cerebellar lobules (*Figure 1J*). These morphological features are not evident in the cerebellum of the mutant mice. In addition to the easily visualized difference in the pattern of rostra-caudal lobules and reduced cerebellar size, the mediolateral morphological divisions of the cerebellum are not apparent and

lamination is almost non-existent in P7 and P14 *En1^Cre/+^;Atoh1^fl/-^* mutant mice (*Figure 1K*). In *En1^Cre/+^;Atoh1^fl/-^* mice, we could identify Purkinje cells (Calbindin^+^ cells, *Figure 1K*), confirming that conditional deletion of *Atoh1* does not block Purkinje cell neurogenesis. However, the cellular organization of Purkinje cells in cerebella from *En1^Cre/+^;Atoh1^fl/-^* mice is markedly different from controls. We confirmed the stunted cerebellar size and lack of lobule complexity by comparing the outline of cerebella from control to *En1^Cre/+^;Atoh1^fl/-^* mice (*Figure 1L*). In line with our conditional knockout that targets neurons in the excitatory lineage, we further found that the representation of different classes of inhibitory neurons, including molecular layer interneurons, basket cells, and Golgi cells, remains robust and clearly detected by cell-type-specific markers (*Figure 1—figure supplement 2E–H*). These data indicate that there are simultaneously occurring alterations in cerebellar morphology and cytoarchitecture in the mutant mice, both owing to the principal defect, which is the elimination of excitatory neurons.

## Excitatory cerebellar neurons guide the transformation of cerebellar clusters into zones

One of the hallmarks of cerebellar circuit organization are the longitudinal zones, or modules, that have a specific pattern and function (*Apps et al., 2018*). While Purkinje cells differentiate into distinct molecular subtypes shortly after birth (*Wassef et al., 1990*), they initially group together into cellular clusters and only form the more precise and sharply delineated longitudinal zones as the cerebellum greatly expands in size and volume (*Dastjerdi et al., 2012*; *Larouche and Hawkes, 2006*). Here, we investigated whether Purkinje cell molecular subtypes in *En1^Cre/+^;Atoh1^fl/-^* mice form distinct zones. As expected, at P14, ZebrinII^+^ and PLCβ4^+^ expressing Purkinje cells in control mice are organized in alternating zones (*Figure 2A–G*). In contrast, ZebrinII^+^ and PLCβ4^+^ expression patterns in the *En1^Cre/+^;Atoh1^fl/-^* cerebellum are organized into a series of clusters rather than sharp zones. The overall pattern of clusters in the mutants resembles the architecture that is typically observed in normal neonates (*Figure 2H–N*; *Fujita et al., 2012*; *Sugihara and Fujita, 2013*). Some Purkinje cells, here marked with PLCβ4^+^, over-migrate beyond the boundaries of the cerebellum and into the inferior colliculus that is immediately adjacent (arrowheads *Figure 2K*). We also observed co-expression of ZebrinII^+^ and PLCβ4^+^ in the same clusters of cells in the *En1^Cre/+^;Atoh1^fl/-^* mice (arrowheads *Figure 2M–N*). In normal mice, postnatal ZebrinII^+^ expression sweeps over the whole cerebellum becoming expressed in all Purkinje cells by around P10, before it segments into zones by ~P15 (*Tano et al., 1992*). The prevalent overlap in what are typically complementary markers at P14 in *En1^Cre/+^;Atoh1^fl/-^* mice supports the hypothesis of an altered temporal progression of cerebellar development that has affected Purkinje cell patterning.

The zonal organization of Purkinje cells is also essential for the formation of patterned synaptic inputs. For example, spinal cord mossy fibers send their projections to the granular layer below ZebrinII^-^ zones (*Sillitoe et al., 2010*). During development, these projections directly target Purkinje cells (*Kalinovsky et al., 2011*; *Sillitoe, 2016*), but with the integration of granule cells into the circuit, mossy fibers displace their synapses onto granule cells (*Arsénio Nunes et al., 1988*; *Pan et al., 2009*). To test how this sequence of events was impacted by the lack of granule cells in *En1^Cre/+^; Atoh1^fl/-^* mice, we injected the anterograde tracer WGA-Alexa 555 into the lower thoracic-upper lumbar spinal cord of P12 mice (*Gebre et al., 2012*; *Lackey and Sillitoe, 2020*). We observed WGA-Alexa 555 labeled fibers and terminals at P14 after two days of tracer transport. In controls, mossy fibers project to the cerebellar cortex in a zonal pattern that reflects the topography of ZebrinII (*Figure 2O–P*; *Brochu et al., 1990*; *Sillitoe and Hawkes, 2002*). As in the controls, we found that spinocerebellar mossy fibers projected mainly to ZebrinII^-^ domains in the *En1^Cre/+^;Atoh1^fl/-^* mice. However, in controls the mossy fibers occupy the domains in the granular layer that are located directly below the Purkinje cell zones, whereas in the mutants the mossy fibers remain intermingled within Purkinje cell clusters (*Figure 2O–Q*). These results confirm that excitatory cerebellar neurons are not essential for initiating the zonal topography of mossy fiber projections that arise from the spinal cord. However, the organization of mossy fibers into poorly defined clusters in the mutants indicates that integration of excitatory neurons into the circuit is important for the architectural rearrangement and the refinement of both the Purkinje cells and afferents into zones.

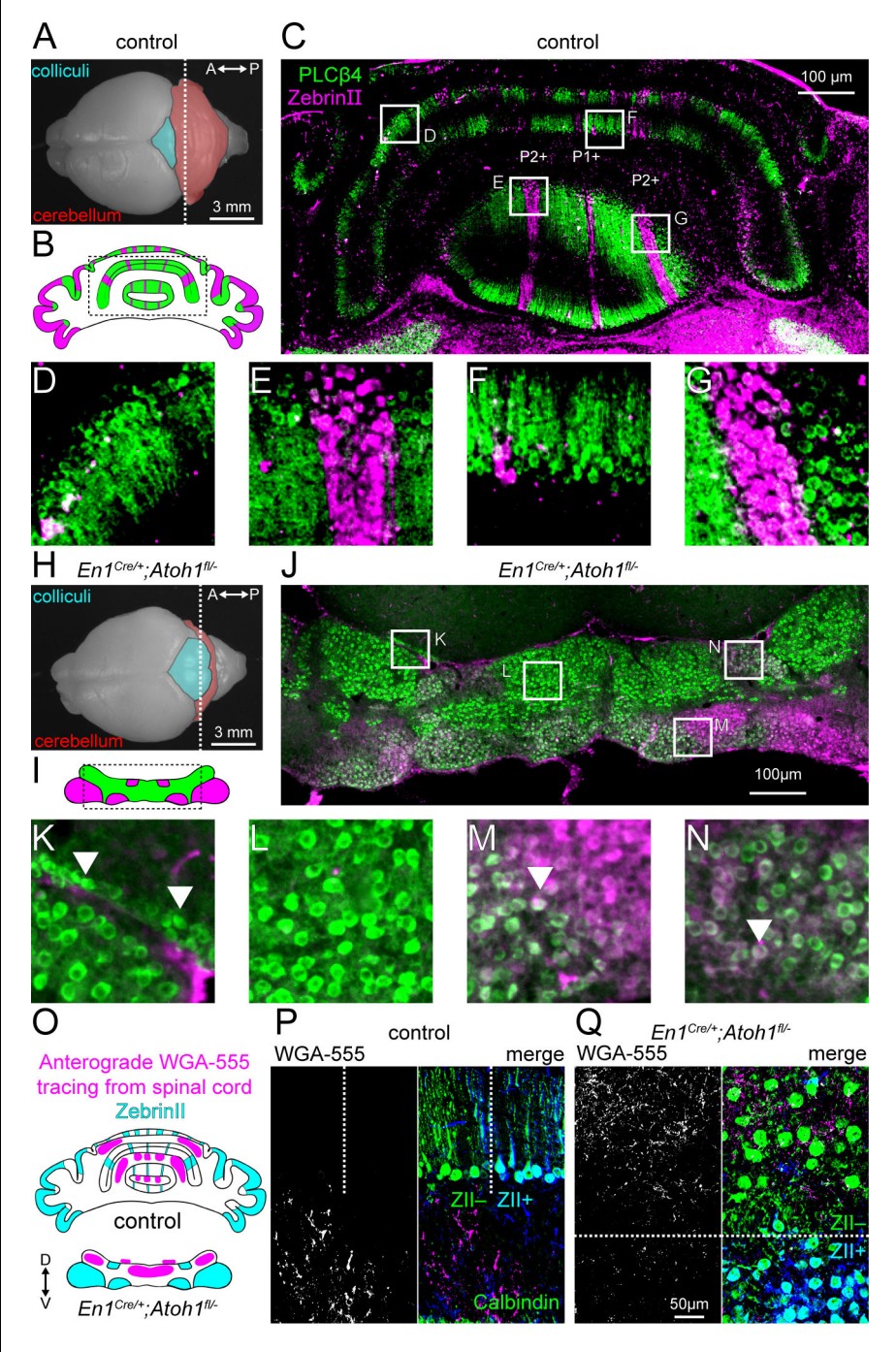

**Figure 2.** *Atoh1* lineage neurons are necessary for the formation of sharp zonal patterns of Purkinje cell sub-types and spinal cord mossy fiber projections. (**A**) Top view of a control P14 brain. Dotted line shows the position of the schematic in (**B**) and where the section in (**C**) was taken from. (**B**) Schematic of Purkinje cell ZebrinII (pink) and PLCβ4 (green) patterns in a control section illustrating the striped patterns at P14. (**C**) Staining of ZebrinII (pink) and PLCβ4 (green). (**D-G**) Higher power magnification images of insets in (**C, H**). Top view of *En1^Cre/+;Atoh1^fl/-* P14 brain. Dotted line shows position of schematic in (**I**) and where the section in (**J**) was taken from. (**I**) Schematic of Purkinje cell ZebrinII (pink) and PLCβ4 (green) patterns in *En1^Cre/+;Atoh1^fl/-* mice showing a clustered pattern at P14. (**J**) Staining of ZebrinII (pink) and PLCβ4 (green). (**K-N**) Higher power magnification images of insets in (**J**) Arrowhead in **K**: over-migrated Purkinje cells in the inferior colliculus. Arrowhead in **M** and **N**: Purkinje cells co-expressing ZebrinII and PLCβ4. (**O**) Schematic of ZebrinII staining pattern and WGA-Alexa 555 tracing form the spinal cord to the cerebellum in control and *En1^Cre/+;Atoh1^fl/-* mice. (**P**) Representative images of WGA-Alexa 555+

*Figure 2 continued*

terminals (left: gray; right: pink) in the cerebellum of a control mouse. Dotted lines represent the border between ZebrinII⁺ (cyan) and ZebrinII⁻ regions. Purkinje cell stained with Calbindin (green) ZebrinII is shown in dark blue. The overlap between Calbindin and ZebrinII is shown cyan. (**Q**) Representative images of WGA-Alexa 555+ terminals (left: gray; right: pink) in the cerebellum of an *En1^{Cre/+};Atoh1^{fl/-}* mouse. Dotted lines represent the border between ZebrinII⁺ (cyan) and ZebrinII⁻ regions. Purkinje cell stained with Calbindin (green). All images are representative of N=3 brains per genotype.

## Excitatory cerebellar neurons promote Purkinje cell dendritic complexity

Previous studies suggest that decreasing excitatory input alters Purkinje cell dendrite outgrowth (*Bradley and Berry, 1976*; *Pan et al., 2009*; *Park et al., 2019*; *Takeo et al., 2021*), which results in

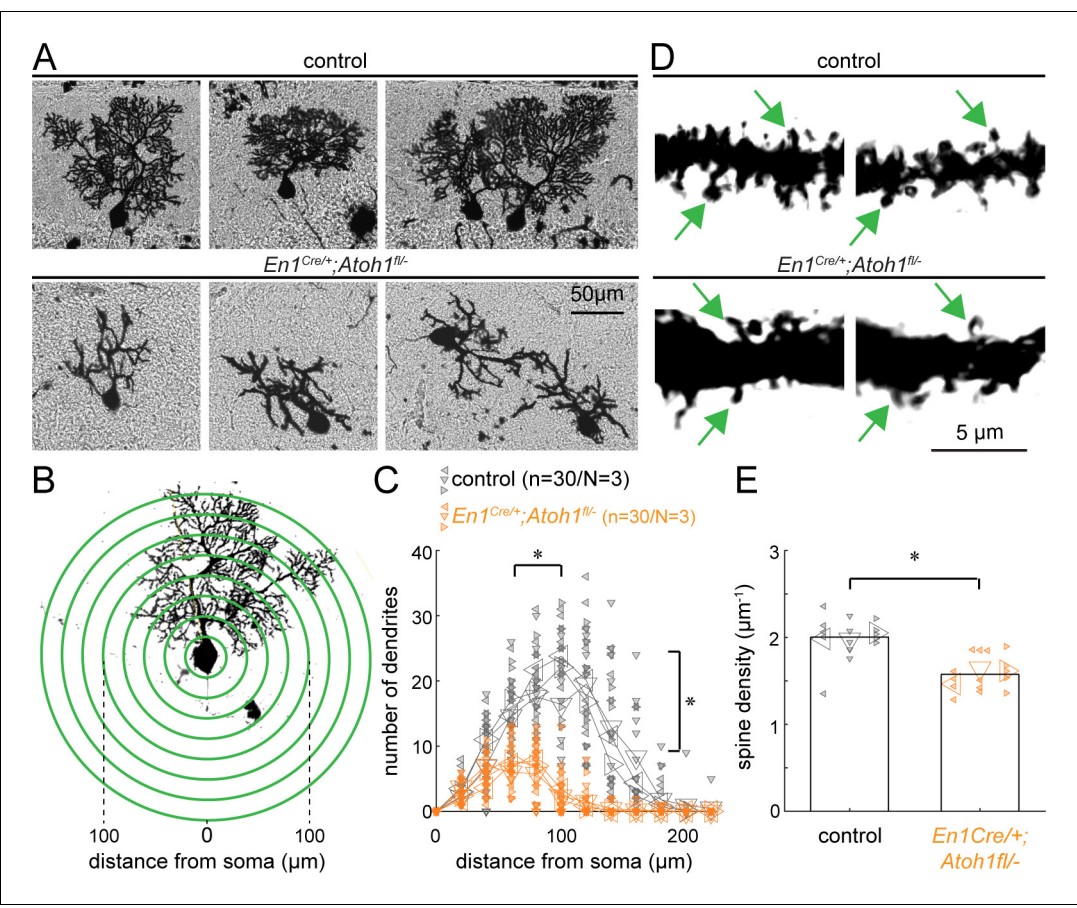

**Figure 3.** *Atoh1* lineage neurons are necessary for establishing the dendritic complexity of Purkinje cells. (**A**) Representative images of Golgi-Cox-labeled Purkinje cells in the cerebella of control (top row) and *En1^{Cre/+}; Atoh1^{fl/-}* (bottom row) mice. (**B**) Sholl analysis for dendritic complexity. (**C**) Purkinje cells in *En1^{Cre/+};Atoh1^{fl/-}* mice have shorter and less branched Purkinje cell dendrites (n=30/N=3 for each genotype, each animal is indicated with a differentially oriented triangle). (**D**) Golgi stains showing dendritic spines for control and *En1^{Cre/+};Atoh1^{fl/-}* mice (examples indicated by green arrowheads). (**E**) *En1^{Cre/+};Atoh1^{fl/-}* mice contain a significantly lower number of dendritic spines compared to control mice. *p<0.001 for spine density (n=15/N=3 for each genotype, each animal is indicated with a differentially oriented triangle). Linear mixed model with genotype as the fixed effect and mouse number as the random effect. *p<0.001 for both the distance from the soma, the branch number, and spine density. All images were acquired from the cerebella of P14 mice. The raw data and specific p-values for the comparisons are presented in *Figure 3—source data 1*.

The online version of this article includes the following source data for figure 3:

**Source data 1.** Source data and specific p-values for representative graphs in *Figure 3*.

abnormal dendrite morphology in agranular mice (*Sotelo and Dusart, 2009*). Therefore, we next investigated the cellular morphology of Purkinje cells in control and *En1^{Cre/+};Atoh1^{fl/-}* mice by staining mouse brains using a modified Golgi-Cox method. We found that Purkinje cells in *En1^{Cre/+}; Atoh1^{fl/-}* mice had stunted and smaller dendritic arbors compared to the controls (*Figure 3A*), and unlike in the control cerebella, neighboring Purkinje cells did not orient their arbors in the same direction (*Figure 3A*). Sholl analysis revealed that the Purkinje cells in the mutants were smaller with less bifurcated dendritic branches (*Figure 3B–C*). Furthermore, spine density was significantly lower in *En1^{Cre/+};Atoh1^{fl/-}* mice compared to the control mice (*Figure 3D–E*), which is in agreement with previous work showing that dendrite outgrowth is dependent on the formation of excitatory synapses, the majority of which are contributed from granule cell parallel fibers in control animals (*Bradley and Berry, 1976*; *Pan et al., 2009*; *Park et al., 2019*; *Takeo et al., 2021*). From these data, we conclude that Purkinje cells in the P14 *En1^{Cre/+};Atoh1^{fl/-}* mice have less morphological complexity. This shows that in addition to causing gross morphological impairments of the cerebellum, loss of *Atoh1* and a reduced number of excitatory cerebellar neurons also negatively affects the cellular level morphology of neurons.

## The cerebellar cortex of *En1^{Cre/+};Atoh1^{fl/-}* mutant mice is largely devoid of Vglut1 synapses

Next, we addressed whether the large reduction of excitatory cerebellar neurons changes the composition of cerebellar cortical afferents. First, we looked how the density of Vglut1⁺ synapses on Purkinje cells changed during the dynamic period. From P7 to P14, a large population of excitatory (Vglut1⁺) granule cells integrate into the cerebellar circuit and make direct contacts to Purkinje cells via their parallel fibers in the molecular layer (*Figure 4A*). While some mossy fibers are also Vglut1⁺, they do not directly contact mature Purkinje cells in control mice but instead target granule cells in the granule cell layer. Therefore, we only quantified the density of Vglut1⁺ synapses in the molecular and Purkinje cell layer. We saw an increase in the density of Vglut1⁺ synapses in the Purkinje cell layer and molecular layer from P7 to P14 in control mice (*Figure 4B*, top row). Because we impaired the neurogenesis of Vglut1-expressing granule cells, we expected the density of Vglut1⁺ synapses to be lower in *En1^{Cre/+};Atoh1^{fl/-}* mice and not increase between P7 and P14. Indeed, we saw that the signal of Vglut1⁺ synapses in *En1^{Cre/+};Atoh1^{fl/-}* mice did not change between P7 and P14 (*Figure 4B*, bottom row) and quantification showed that this signal was significantly lower at both time points in *En1^{Cre/+};Atoh1^{fl/-}* mice compared to age-matched control mice (*Figure 4C*). The remaining Vglut1⁺ synapses in the *En1^{Cre/+};Atoh1^{fl/-}* mice likely originate from mossy fibers (see *Figure 2*), and the small number of granule cells and unipolar brush cells that escaped our genetic manipulation of rRL-derived cells (*Figure 1—figure supplement 2*, *Figure 1—figure supplement 3*, and *Figure 1—figure supplement 4*). However, the approximately nine-fold reduction in Vglut1⁺ signal in the cerebella of *En1^{Cre/+};Atoh1^{fl/-}* mice further confirms that our conditional knockout strategy impairs the integration of granule cells into the cerebellar circuit. Taken together, we show that without excitatory cerebellar neurons, the density of Vglut1⁺ excitatory synapses in *En1^{Cre/+};Atoh1^{fl/-}* mice is much lower when assessed on individual Purkinje cells (*Figure 3D–E*) as well as throughout the morphogenetically abnormal mutant cerebellum at large (*Figure 4C*).

## Excitatory cerebellar neurons shape Purkinje cell simple spike firing patterns

Purkinje cells have a distinct firing profile characterized by spontaneously generated simple spikes and climbing fiber-induced complex spikes. In vitro recordings in rats (*Crepel, 1972*; *McKay and Turner, 2005*) and mice (*Beekhof et al., 2021*) show that Purkinje cell firing properties change significantly during early postnatal development (P7-P14), but it is unclear how firing patterns evolve in vivo during this dynamic period of rewiring. Therefore, we set out to answer two questions: first, how do the in vivo Purkinje cell firing patterns change during the dynamic period? Second, is Purkinje cell firing affected in *En1^{Cre/+};Atoh1^{fl/-}* mice that have a reduced number of Vglut1⁺ excitatory inputs? We hypothesized that the reduced number of Vglut1⁺ excitatory inputs affects the modulation of simple spike firing rate. Based on the time points we tested, and in line with the highest Vglut1⁺ density in P14 control mice, Purkinje cells recorded from these mice had the highest peak firing frequency, whereas Purkinje cells in P7 control mice had an overall lower firing rate

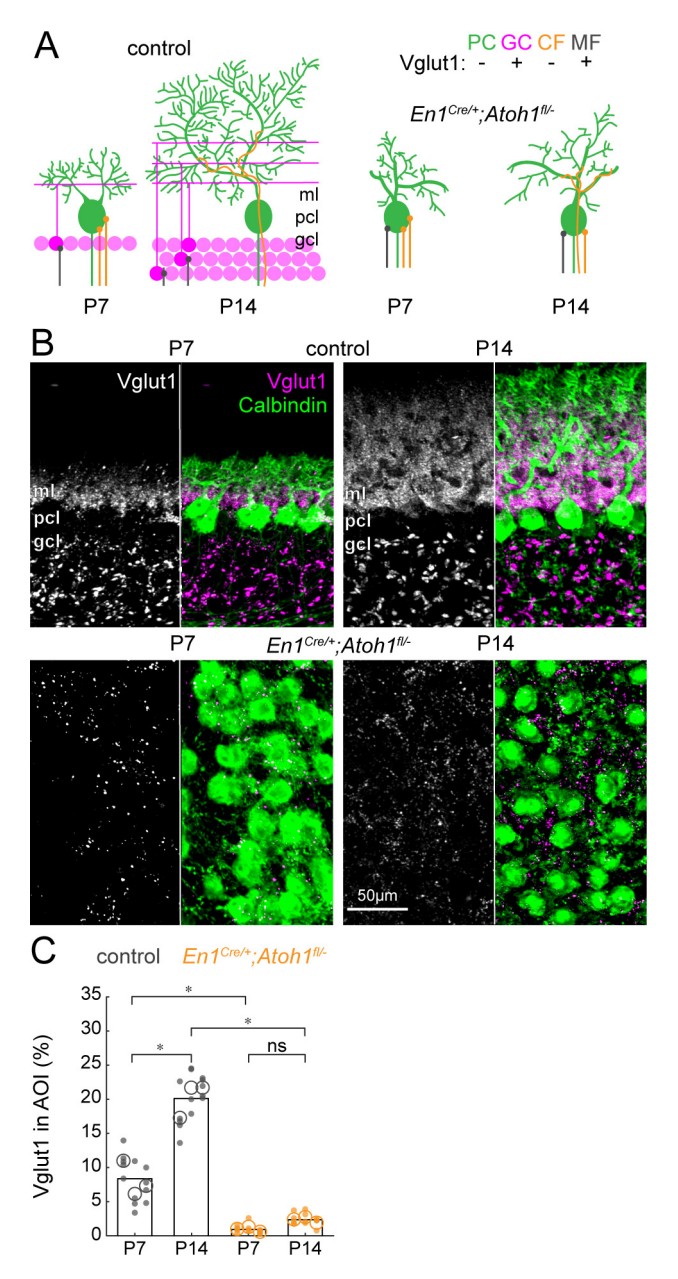

**Figure 4.** The density of Vglut1 synapses onto Purkinje cells increases between P7 and P14 in control mice and is lower in *En1^{Cre/+};Atoh1^{fl/-}* mice. (**A**) Schematic of Purkinje cell microcircuit in control (left) and *En1^{Cre/+};Atoh1^{fl/-}* mice (right) at P7 and P14. Vglut1 is expressed in granule cell (pink) and some mossy fiber (gray) terminals but not in Purkinje cells (green) or climbing fiber (orange) terminals. Abbreviations: ml = molecular layer; pcl = Purkinje cell layer; gcl = granule cell layer. (**B**) Representative images for Vglut1 density in the cerebellum of control (top) and *En1^{Cre/+};Atoh1^{fl/-}* mice (bottom) at P7 and P14. Left panels are gray-scale images of the Vglut1 signal and right images show the merged images of the Vglut1 (pink) and Calbindin (green) signals. (**C**) Density of Vglut1 synapses as a percentage of area of interest (AOI; AOI comprises the molecular and Purkinje cell layer in cerebella from control animals and the area occupied by Purkinje cells in the cerebella of *En1^{Cre/+};Atoh1^{fl/-}* mice). Significance was established based on the average of each mouse (N=3 mice per genotype, large circles; n=3–5 tissue sections, small dots) using a two-way ANOVA (genotype*age) followed by a Tukey Kramer post-hoc analysis. *p=0.05. The raw data and specific P-values for the comparisons are presented in *Figure 4—source data 1*. The online version of this article includes the following source data for figure 4:

**Source data 1.** Source data and specific p-values for representative graphs in *Figure 4*.

(*Figure 5A–B*). Interestingly, Purkinje cells in P14 controls fired in burst-like patterns that have unique powerfully represented features in their firing frequency and frequency mode (or preferred frequency) (*Figure 5B and D*), whereas Purkinje cells in P7 control mice fired more continuously, yet at a lower frequency and with higher variability in their inter-spike intervals (*Figure 5A and E*). We quantified the firing features with six parameters: frequency (total spikes/recording time; *Figure 5F*), frequency mode (most frequently observed frequency; *Figure 5G*), CV (a measure for global regularity; *Figure 5J*), pause percentage (defined as a discrete portion of the trace where no spikes were observed; *Figure 5K*), rhythmicity index (the oscillatory properties in the auto-correlation of spike times; *Figure 5L–N*), and CV2 (inter-spike interval irregularity; *Figure 5O*; *Holt et al., 1996*). Specifically, a faster firing pattern is represented as a higher frequency and frequency mode (*Figure 5D–G*). A more burst-like, or globally irregular, firing pattern (*Figure 5H–I*) can be observed as a larger difference between the observed frequency and frequency mode (*Figure 5D–G*), a higher CV (*Figure 5J*), and a larger pause proportion (*Figure 5K*). Additionally, the higher inter-spike interval (local) irregularity is reflected as a lower rhythmicity index (*Figure 5L–N*) and higher CV2 (*Figure 5O*).

We calculated these parameters for Purkinje cells recorded from control mice P7-8, P9-10, P11-12, and P13-14. We found that there is a statistically significant increase in frequency and frequency mode from P7-10 to P11-14 (*Figure 5F–G*). We also found a statistically significant increase in CV and pause proportion from P7-12 to P13-14 (*Figure 5J–K*). In addition, there was an increase in the rhythmicity index from P7-10 to P11-14 (*Figure 5N*), but no change in the CV2 (*Figure 5O*). Together, these results suggest that there are changes in the firing patterns that occur between P7-10 and P13-14, and that these changes in activity involve a higher firing frequency, with more burst-like firing patterns, and a higher rhythmicity within the burst. The changes in firing patterns are further visualized in the firing frequency distributions for each single cell that we recorded for this study (*Figure 5—figure supplement 1*).

Next, we set out to quantify the firing patterns in $En1^{Cre/+};Atoh1^{fl/-}$ mutants (*Figure 5C*). We decided to measure spontaneous Purkinje cell signals at P10 and P14, because P9-10 control cells were never statistically different from P7-8 in any of the parameters included in our study and because P13-14 control cells were different from P7-10 control cells in five out of six parameters we included. We found that spontaneous simple spikes in Purkinje cells recorded in P10 and P14 $En1^{Cre/+};Atoh1^{fl/-}$ mutants had a lower frequency and frequency mode than those recorded in Purkinje cells in P11-14 control mice (*Figure 5D–G*). We also found that the CV and pause proportion was significantly lower in Purkinje cells recorded from P10 and P14 $En1^{Cre/+};Atoh1^{fl/-}$ mutants compared to the P13-14 control Purkinje cells (*Figure 5H–K*). In addition, we found that Purkinje cells recorded from P10 and P14 $En1^{Cre/+};Atoh1^{fl/-}$ mutants had a lower rhythmicity index than the P11-14 control Purkinje cells (*Figure 5L–N*) and a higher CV2 than all control Purkinje cells (*Figure 5O*). Finally, we found that Purkinje cell firing patterns in P10 and P14 $En1^{Cre/+};Atoh1^{fl/-}$ mutants never differed from each other in any of the six parameters that we included in our analysis. This observation is further supported by the similarity in firing frequency distributions of the single Purkinje cell recordings in P10 and in P14 in $En1^{Cre/+};Atoh1^{fl/-}$ mutants (*Figure 5—figure supplement 1*). Together, these results suggest that the changes in Purkinje firing patterns that occur between P7-10 and P13-14 in control animals does not occur in $En1^{Cre/+};Atoh1^{fl/-}$ mutants, and that Purkinje cells in P10 and in P14 in $En1^{Cre/+};Atoh1^{fl/-}$ mutants have a firing pattern that is highly similar to Purkinje cells in the P7-10 control mice.

## An electrophysiological pseudo-timeline uncovers a developmental transformation in Purkinje cell firing properties

Based on the six parameters quantified in *Figure 5*, P14 Purkinje cells in $En1^{Cre/+};Atoh1^{fl/-}$ mice appear more dissimilar from their age-matched controls than younger Purkinje cells in control mice. This is further supported when looking at representative traces from Purkinje cells recorded in P7 to P14 control mice and P10 and P14 $En1^{Cre/+};Atoh1^{fl/-}$ mice (*Figure 6A*). However, no single parameter alone can fully describe the complex temporal dynamics of the transformation that occurs in Purkinje cell firing patterns during the second postnatal week. Therefore, we used several mathematical approaches to compare the firing patterns of the 149 single neuron recordings, based on all six firing properties described in *Figure 5*. First, we performed an unbiased cluster analysis that groups together neurons that have the most similar firing properties (*Figure 6B*). When we retrospectively

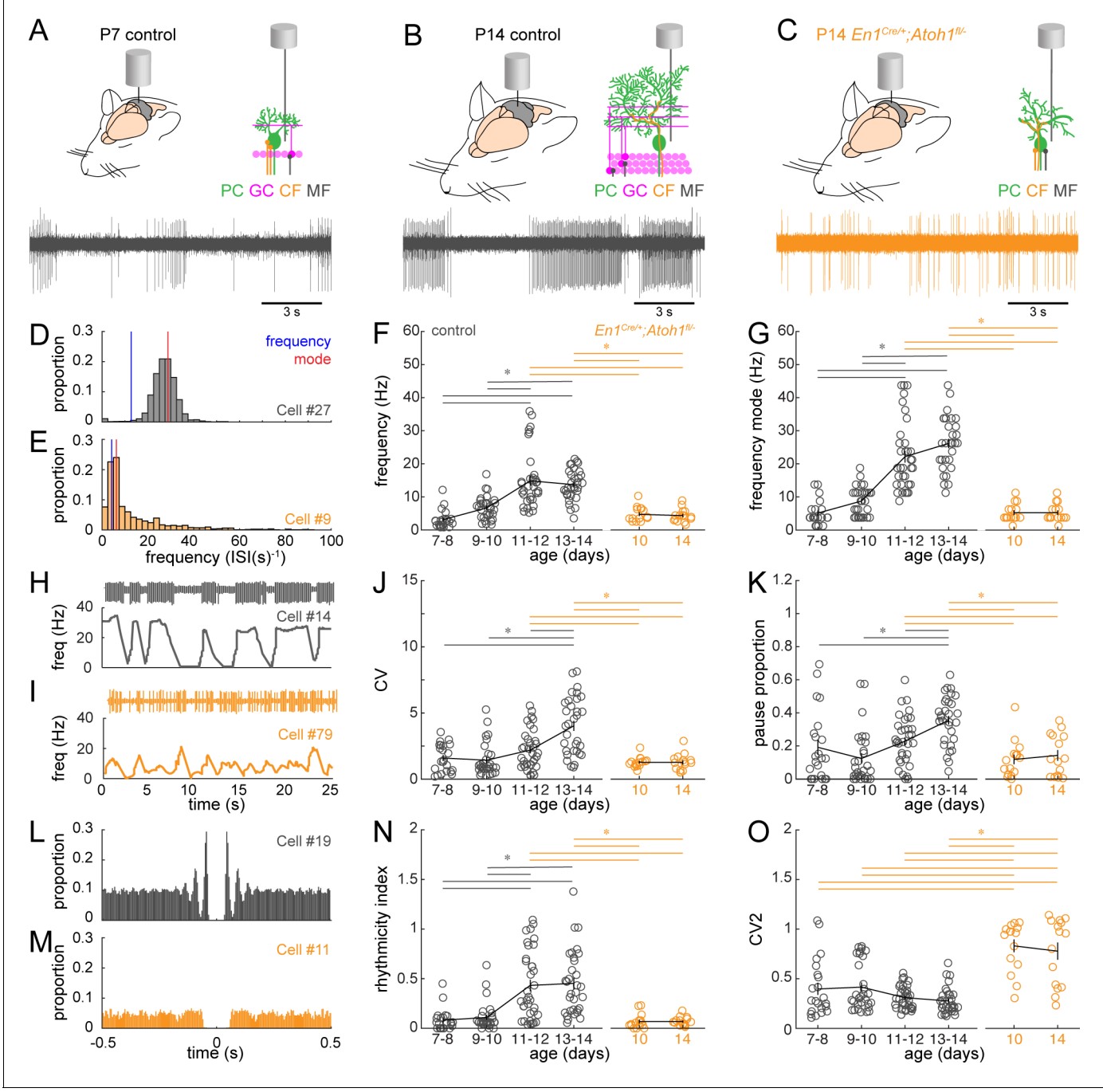

**Figure 5.** Purkinje cell simple spike firing patterns transform from P7 to P14 in control mice and reflect altered maturation in P14 *En1^{Cre/+};Atoh1^{fl/-}* mice. (**A**) Schematic of in vivo Purkinje cell recordings in P7 control mouse. Top left: cerebellum with recording electrode. Top right: schematic with recording electrode extracellular to Purkinje cell body. Bottom: representative 15 s recording from a Purkinje cell, each large vertical line is an action potential. (**B**) Same as (**A**) For P14 control mouse. (**C**) Same as (**A**) for P14 *En1^{Cre/+};Atoh1^{fl/-}* mouse. (**D**) and (**E**) Examples of frequency (interspike interval, $ISI^{-1}$) distributions of simple spike firing rate in a P14 Purkinje cell recorded in a control (**D**) and *En1^{Cre/+};Atoh1^{fl/-}* (**E**) mouse. Blue line indicates the 'frequency' as calculate by the total number of simple spikes/recording time (spikes/s). Red lines indicate 'frequency mode' as the frequency most commonly observed in the frequency distribution. (**F**) Simple spike firing frequency (spikes/recording time). (**G**) Simple spike frequency mode (peak $ISI^{-1}$ distribution). (**H**) and (**I**) example recordings (25 s) from P14 Purkinje cells in control (**H**) and *En1^{Cre/+};Atoh1^{fl/-}* (**I**) mouse. The top is the recording. Bottom is the firing frequency averaged over one second. (**J**) Simple spike CV (global firing irregularity). (**K**) Pause percentage (proportion of recording with ISI > five times average ISI). (**L**) and (**M**) Examples of auto-correlograms of the simple spike ISI in a P14 Purkinje cell recorded in a control (**L**) and *En1^{Cre/+}; Atoh1^{fl/-}* (**M**) mouse. (**N**) Simple spike rhythmicity index calculated based on auto-correlogram. (**O**) Simple spike CV2 (local firing irregularity). For **F,G, J, K, N**, and **O**, significance was determined using an ANOVA between the four control age groups and two *En1^{Cre/+};Atoh1^{fl/-}* age groups followed by a

*Figure 5 continued on next page*

*Figure 5 continued*

Tukey-Kramer post-hoc analysis to assess differences between the groups. Significance was accepted at p<0.05. In the figure, the statistical significance between two groups is indicated as a line starting and ending above the two groups. Gray lines indicate differences within control groups of different ages and orange lines indicate differences between control and *En1$^{Cre/+}$;Atoh1$^{fl/-}$* groups. N/n-numbers: control: P7-8: n=22 cells from N=13 mice; control P9-10: n=30/N=12; control P11-12: n=35/N=11; control P13-14: n=32/N=14; *En1$^{Cre/+}$;Atoh1$^{fl/-}$* P10: n=15/N=3; *En1$^{Cre/+}$;Atoh1$^{fl/-}$* P14: n=15/N=6. The raw data and specific p-values for all the comparisons are presented in *Figure 5—source data 1*.

The online version of this article includes the following source data and figure supplement(s) for figure 5:

**Source data 1.** Source data and specific p-values for representative graphs in *Figures 5* and *6*.
**Figure supplement 1.** Simple spike distributions of single Purkinje cells.

label the age and genotype of each cell after unbiased clustering, we see that older control cells cluster together (P11-14 in difference shades of green, clustered at the top) and younger control cells cluster together (P7-10 in different shades of purple, cluster at the bottom). Interestingly, Purkinje cells from both P10 and P14 *En1$^{Cre/+}$;Atoh1$^{fl/-}$* mice cluster with the young control cells (orange, at the bottom) (*Figure 6B*). Second, we performed t-distributed stochastic neighbor embedding (t-SNE) analysis on the aforementioned firing parameters (*Figure 6C*). After retrospectively color-coding the data points, we present a pseudo-timeline of the developmental transformation of firing patterns, with Purkinje cells from both P10 and P14 *En1$^{Cre/+}$;Atoh1$^{fl/-}$* mice clustered with the young (P7-10) control cells in the bottom right and older (P11-14) control cells on clustered together on the top left (*Figure 6C*). Both the unbiased cluster and the tSNE analysis show some cells in an intermediate state (between young and old) (*Figure 6B–C*, middle cells). This suggests the existence of a transformation that occurs between younger and older cells, but the transformation does not occur at the same time for all Purkinje cells, as is expected based on previous studies showing that the heterogeneity of Purkinje cell function is an inherent property of Purkinje cells that respects and spans the timelines of anatomical and functional development of Purkinje cells (*Beekhof et al., 2021*; *McKay and Turner, 2005*; *van Welie et al., 2011*). To further test whether there are general changes that occur at different time points or a transformation that occurs at a discrete age, we averaged the firing properties of all cells recorded in each group, allowing us to test for the relative contribution of specific firing properties in each age group, and performed an unbiased, hierarchical cluster analysis on the first two principal components of the averaged firing properties (*Figure 6D*). We find that although the young control cells (P7-10) are most similar to themselves and most dissimilar from older control cells (P11-14), there is a level of variability that likely reflects systematic differences for when the transformation occurs across different Purkinje cells. In agreement with our aforementioned findings, we found that the P10 and P14 Purkinje cells recorded in *En1$^{Cre/+}$;Atoh1$^{fl/-}$* mice cluster with the young control cells (P7-10) and do not undergo the functional transformation in firing patterns that Purkinje cells in control mice do. Taken together, in a period of dynamic circuit development, Purkinje cells undergo a major developmental transformation in firing patterns between P7-10 and P11-14. Furthermore, in agreement with the absence of change in gross cerebellar morphology or dynamics in Vglut1$^+$ synapses, this developmental transformation does not occur between P10 and P14 in *En1$^{Cre/+}$;Atoh1$^{fl/-}$* mice.

Next, we tested whether the altered neural activity reflects a delay in maturation of the firing patterns. We therefore recorded Purkinje cells in P18 control and *En1$^{Cre/+}$;Atoh1$^{fl/-}$* mice, which is a week after the changes in the electrophysiological signature occur in control mice and the latest timepoint at which we are confident that the general health of the *En1$^{Cre/+}$;Atoh1$^{fl/-}$* mice does not confound the quality of our recordings. We find that at P18, Purkinje cells recorded in *En1$^{Cre/+}$; Atoh1$^{fl/-}$* mice still exhibit a lower firing frequency (*Figure 6—figure supplement 1A*) and preferred firing frequency (*Figure 6—figure supplement 1A*), (*Figure 6—figure supplement 1B*), no difference in pause proportion (*Figure 6—figure supplement 1C*), lower rhythmicity index (*Figure 6—figure supplement 1D*), lower global firing rate variability (*Figure 6—figure supplement 1E*), and higher local firing rate variability (*Figure 6—figure supplement 1F*) compared to controls. We then reran the unbiased clustering analysis and tSNE analysis on all 179 single-cell recordings, and the hierarchical cluster analysis on the all group means to investigate whether P18 Purkinje cells in control and *En1$^{Cre/+}$;Atoh1$^{fl/-}$* mice more closely relate to Purkinje cells at the start (P7-10) or end (P11-14) of the dynamic developmental period. We found that Purkinje cells recorded in P18 control mice cluster with Purkinje cells recorded in P11-14 control mice, but Purkinje cells recorded in P18

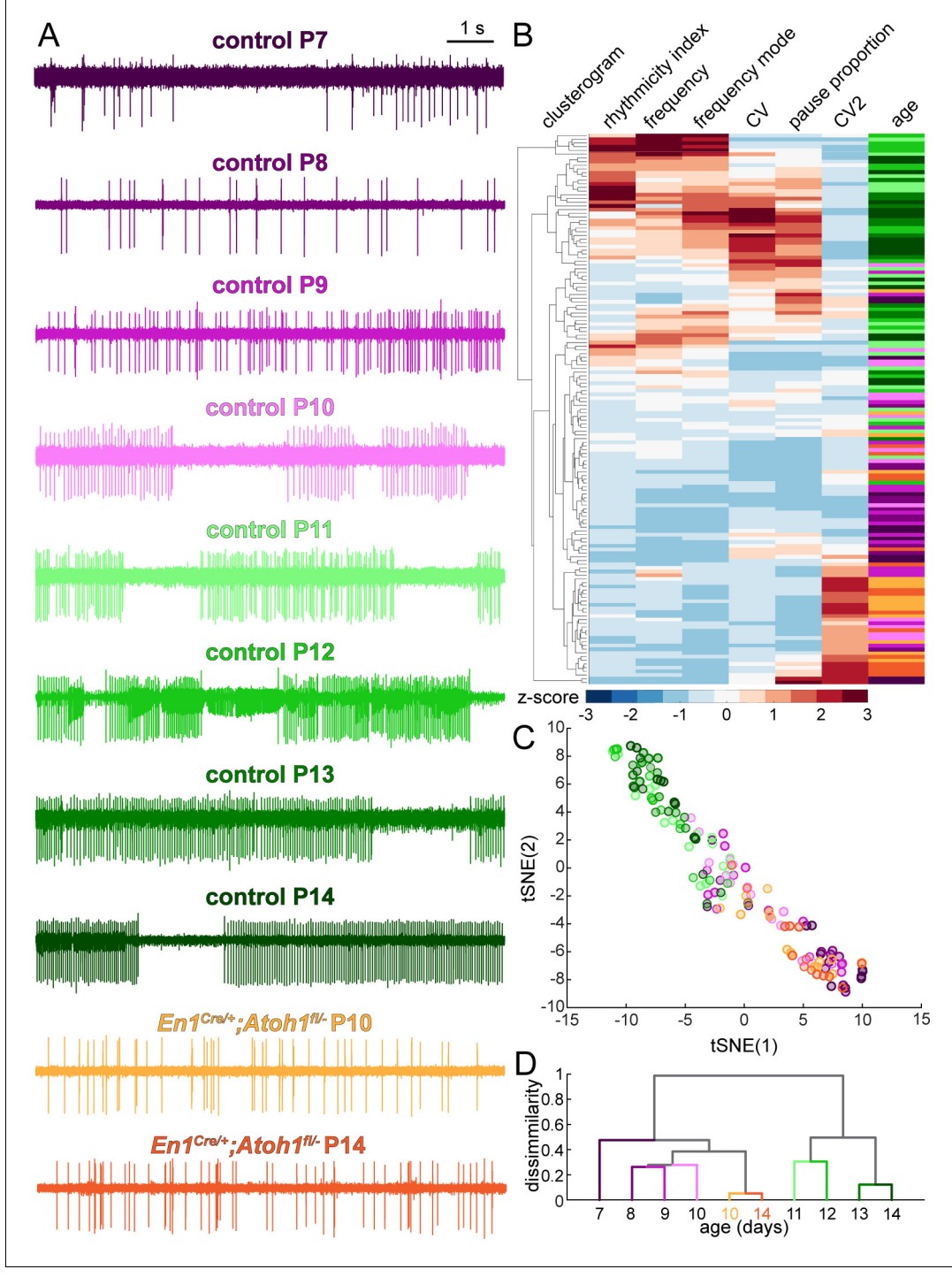

**Figure 6.** Purkinje cell simple spike firing patterns undergo a developmental transformation between P7-P10 and P11-P14 that does not occur in *En1^Cre/+^;Atoh1^fl/-^* mice. (**A**) Representative electrophysiological recordings (each 10 s long) from Purkinje cells recorded in P7-P14 control mice and P10 and P14 *En1^Cre/+^;Atoh1^fl/-^* mice. In some recordings, we observed action potentials from a second cells in the background, typically observed as spikes with a smaller amplitude (for example, in the representative P12 trace). We only include the large amplitude spikes of the single predominant cell in our analysis. (**B**) Unbiased clusterogram based on the firing parameters summarized in *Figure 5*. Each row represents the Z-scored firing parameters of a single neuron. The rows were retroactively color-coded according to the colors in panel **A**. (**C**) tSNE analysis of the firing patterns summarized in *Figure 5*. (**D**) Unbiased cluster analysis on average firing patterns of Purkinje cells recorded in P7-P14 control mice and P10 and P14 *En1^Cre/+^;Atoh1^fl/-^* mice.

*Figure 6 continued on next page*

*Figure 6 continued*

The online version of this article includes the following figure supplement(s) for figure 6:

**Figure supplement 1.** Simple spike firing properties do not normalize by P18 in *En1$^{Cre/+}$;Atoh1$^{fl/-}$* mice.

*En1$^{Cre/+}$;Atoh1$^{fl/-}$* mice cluster with the younger Purkinje cells that were recorded at P7-10 in control mice, which were also similar to Purkinje cells recorded in P10 and P14 *En1$^{Cre/+}$;Atoh1$^{fl/-}$* mice (*Figure 6—figure supplement 1G–I*). These analyses further underscore that developing Purkinje cells rely on the neurogenesis of their surrounding excitatory neurons to make a dynamic functional transformation in their firing properties and that Purkinje cells developing in an environment largely devoid of granule cells are maintained in a functionally immature state, rather than acquiring a new, pathophysiological firing pattern.

## Excitatory cerebellar neurons are necessary for pruning away excess Vglut2 synapses from Purkinje cells

Our results argue that excitatory neurons are essential for shaping the spontaneous firing properties of Purkinje cells. However, a subset of Purkinje cell action potentials is dynamically induced by excitatory inputs from climbing fibers originating in the inferior olive. Interestingly, during early postnatal development, multiple climbing fibers contact one Purkinje cell, but in a parallel-fiber/granule cell-dependent process, the additional climbing fibers are pruned away during the second and third weeks of life (*Crepel and Delhaye-Bouchaud, 1979*; *Hashimoto et al., 2009*; *Kano and Hashimoto, 2012*). Immature Vglut2$^+$/climbing fiber synapses are located around the Purkinje cell body in controls, but at around P9 only the 'winner' among these synapses translocates to the Purkinje cell dendrites in the molecular layer (*Kano et al., 2018*). In addition to this change in Vglut2$^+$-expressing climbing fibers, mossy fibers temporarily innervate Purkinje cells bodies (*Kalinovsky et al., 2011*; *Sillitoe, 2016*) and granule cells may transiently express Vglut2 (*Miyazaki et al., 2003*). We therefore expected a reduction in Vglut2$^+$-positive synapses around the Purkinje cell (*Figure 7A*). Indeed, we observed a decrease in the density of Vglut2$^+$ synapses in P14 control cerebella compared to P7, likely due to a combined reduction in Vglut2$^+$ expression in granule cells, displacement of mossy fibers to granule cells, and pruning of climbing fibers (*Figure 7B–C*). At both timepoints, however, we observed a high density of Vglut2$^+$ synapses in the cerebella of *En1$^{Cre/+}$;Atoh1$^{fl/-}$* mice (*Figure 7B–C*). Given the large reduction in the total number of granule cells (*Figure 1—figure supplement 3*), it is likely that this high density of Vglut2$^+$ synapses stems from mossy fibers and climbing fibers directly innervating Purkinje cell bodies. We tested this hypothesis by counting the number of Purkinje cells that had Vglut2$^+$ puncta on their cell bodies. As expected, the number of soma-associated Vglut2$^+$ contacts on Purkinje cells decreased between P7 and P14 in control mice, whereas all Purkinje cell somata in the *En1$^{Cre/+}$;Atoh1$^{fl/-}$* mice remained enriched with Vglut2$^+$ synapses (*Figure 7D*). These data show that the *En1;Atoh1* lineage, excitatory cerebellar neurons are necessary for shaping the regional localization and cellular targeting of immature mossy fibers and climbing fibers that project to the Purkinje cells.

## Loss of excitatory cerebellar neurons alters Purkinje cell complex spike firing patterns

Based on our findings that *En1;Atoh1* lineage excitatory cerebellar neurons are necessary for the architectural maturation of mossy fibers and climbing fibers, we next wanted to investigate how climbing fiber-induced complex spikes changed between P7 and P14 control mice and if they were altered in *En1$^{Cre/+}$;Atoh1$^{fl/-}$* mice (*Figure 8A–C*). Interestingly, the predominant shape of the complex spike was different between P7 and P14. Purkinje cells in P14 control mice fired classical complex spikes, which are defined by a large sodium spike followed by a train of three to five smaller spikelets (*Davie et al., 2008*; *Zagha et al., 2008*; *Figure 8B*). In contrast, Purkinje cells in P7 control mice fire 'doublets', which are characterized by an initial simple spike-like action potential, followed by a smaller action potential that occurs within 20 ms (a similar profile of doublets was previously reported in neonatal rats [*Puro and Woodward, 1977*; *Sokoloff et al., 2015*, *Figure 8A*]). Similar to Purkinje cells in P7 control mice, the Purkinje cells in P10 and P14 *En1$^{Cre/+}$;Atoh1$^{fl/-}$* mice also predominantly fired doublets instead of true complex spikes, suggesting that the extrinsically-induced

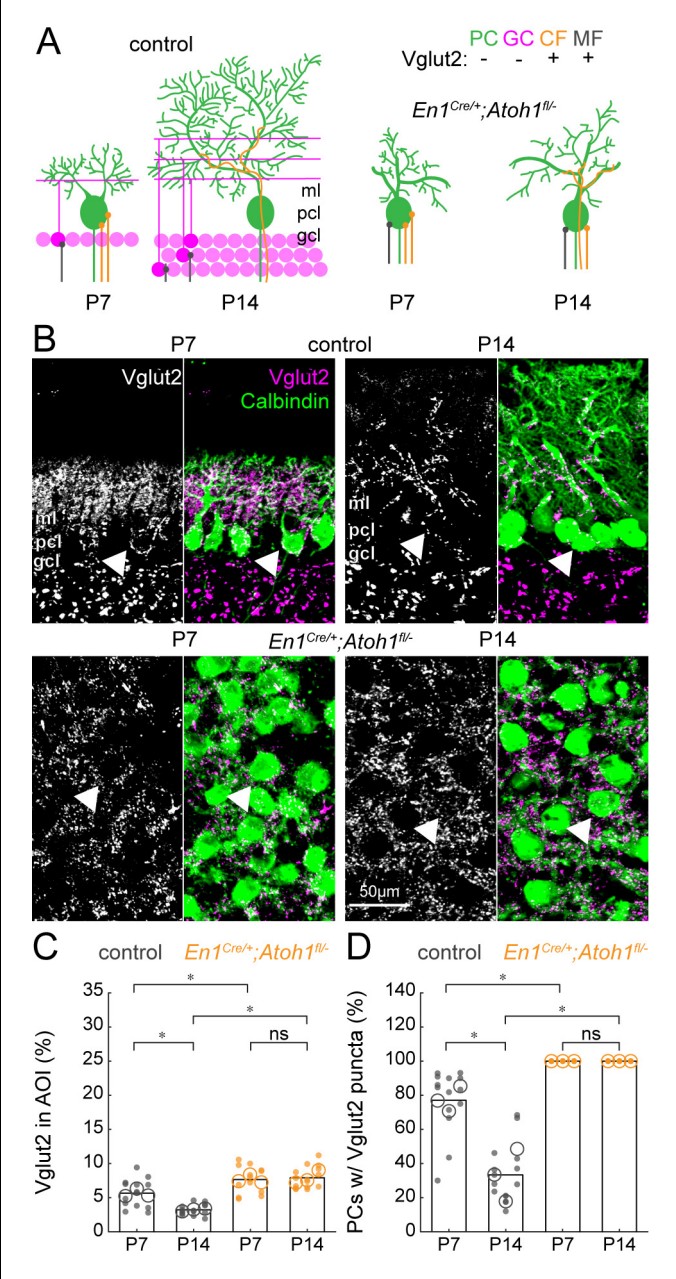

**Figure 7.** The density of Vglut2 synapses onto Purkinje cells decreases between P7 and P14 in control mice and is higher in *En1^{Cre/+};Atoh1^{fl/-}* mutants. (**A**) Schematic of Purkinje cell microcircuit in control (left) and *En1^{Cre/+};Atoh1^{fl/-}* mice (right) at P7 and P14. Vglut2 is expressed in climbing fibers (orange) and some mossy fibers (gray) but not in Purkinje cells (green) or granule cell (pink) terminals at P14, but in some granule cells at P7 (*Miyazaki et al., 2003*). Abbreviations: ml = molecular layer; pcl = Purkinje cell layer; gcl = granule cell layer. (**B**) Representative images showing the density of Vglut2⁺ profiles in the cerebella of control (top) and *En1^{Cre/+};Atoh1^{fl/-}* mice (bottom) at P7 and P14. Left panels are gray-scale images of the Vglut1 signal and right panels are merged to show the Vglut2 (pink) and Calbindin (green) signals. Arrowheads point to Purkinje cells that have direct Vglut2 synapses on their cell bodies. (**C**) Density of Vglut2 synapses as a percentage of the area of interest (AOI; AOI comprises the molecular and Purkinje cell layer in cerebella from control animals and the area occupied by Purkinje cells in the cerebella of *En1^{Cre/+};Atoh1^{fl/-}* mice). (**D**) Number of Purkinje cells with direct Vglut2⁺ puncta on their cell bodies. Significance in **C** and **D** was established based on the average of each mouse (N=3 mice per genotype, large circles; n=3–5 tissue sections, small dots) using a two-way ANOVA (genotype*age) followed by a Tukey Kramer post-hoc analysis. *p=0.05. The raw data and specific p-values for all the comparisons are presented in *Figure 7— source data 1*.

*Figure 7 continued on next page*

Figure 7 continued

The online version of this article includes the following source data for figure 7:

**Source data 1.** Source data and specific p-values for representative graphs in *Figure 7*.

action potentials in Purkinje cells are maintained in an immature state just like the firing patterns of spontaneous simple spikes (*Figures 5* and *6*). Furthermore, we found that some Purkinje cells fired both complex spikes and doublets, which may indicate two separate climbing fibers promoting distinct types of post-synaptic spikes (example *Figure 8D*). We observed heterogeneity in complex spike type in about half of the Purkinje cells recorded in the youngest P7-10 control mice (P7-8: 10/21 cells; P9-10: 19/30 cells) and in about one third of Purkinje cells in the older P11-14 control mice (P11-12: 9/35 cells; P13-14: 12/32 cells). However, the majority of Purkinje cells recorded from P10 $En1^{Cre/+};Atoh1^{fl/-}$ mice (P10: 12/15 cells) and all Purkinje cells recorded from P14 $En1^{Cre/+};Atoh1^{fl/-}$ mice (P14: 15/15 cells) showed complex spike heterogeneity. These data may collectively indicate a heterogeneity in climbing fiber innervation that is reduced within the second postnatal week and dependent on the integration of excitatory neurons within the cerebellar circuit.

Another interesting observation we made was the occurrence of 'complex spike trains', which we defined as three or more complex spikes that followed each other in close succession (*Figure 8E*). These complex spike trains were observed in Purkinje cells recorded in control mice of all ages (P7-8: 13/22 cells; P9-10: 8/30 cells; P11-12: 8/35 cells; P13-14: 9/32 cells) as well as $En1^{Cre/+};Atoh1^{fl/-}$ mice (P10: 5/15; P14: 15/15 cells) and may be a necessary property that helps establish, stabilize, and/or maintain inferior olive to Purkinje cell climbing fiber synapses.

When we quantified the number of different types of complex spikes, we found that classical complex spikes occurred less often in Purkinje cells of young control animals and $En1^{Cre/+};Atoh1^{fl/-}$ mutants. Specifically, we found that Purkinje cells recorded in P11-14 control mice had a statistically significant higher frequency of complex spikes than Purkinje cells recorded from P7 to P10 control mice and P10 to P14 $En1^{Cre/+};Atoh1^{fl/-}$ mutants (*Figure 8F*). Conversely, the frequency of doublets was significantly higher in Purkinje cells recorded from P10 to P14 $En1^{Cre/+};Atoh1^{fl/-}$ mutants than in Purkinje cells in control animals of all ages (*Figure 8G*). In addition, the number of all combined complex spikes was higher in Purkinje cells recorded from in P14 $En1^{Cre/+};Atoh1^{fl/-}$ mutants than that observed in Purkinje cells recorded in P7-8, P9-10, and P13-14 control animals (*Figure 8H*). Together, these results indicate that the Purkinje cells in the mutant mice do not acquire a similar maturation in overall complex spike shape that the control Purkinje cells do and they have a high number of immature-like doublets, which may indicate an altered pruning process of immature climbing fibers.

## The proper establishment of cerebellar-dependent behaviors requires the presence of a full repertoire of excitatory cerebellar neurons

Finally, we wanted to investigate how the observed changes in cerebellar function are translated into behavioral abnormalities. Purkinje cells send projections to the cerebellar nuclei, which consists of both excitatory and inhibitory neurons, as well as some direct projections to the vestibular nuclei (*Sillitoe and Joyner, 2007*). These circuits are important for motor coordination and balance as well as social behaviors including ultrasonic vocalization (USV) in neonatal pups (*Fujita et al., 2008*; *Lalonde and Strazielle, 2015*). Interestingly, the contribution of the cerebellum to these behaviors is initiated before circuit rewiring is completed (*Lalonde and Strazielle, 2015*). Therefore, we were curious to know whether the immature circuit of $En1^{Cre/+};Atoh1^{fl/-}$ mice was sufficient for performing a substantial repertoire of cerebellar-dependent behaviors. Observations of control and $En1^{Cre/+};Atoh1^{fl/-}$ mutant mice showed overt phenotypic differences in motor control (*Video 1* and *Figure 9A*) similar to what was previously observed in other agranular mice (*Sidman et al., 1962*; *Sidman et al., 1965*; *Woodward et al., 1974*). At P14, control mice explore an open arena with smooth intentional motions, whereas $En1^{Cre/+};Atoh1^{fl/-}$ mice often fall on their backs. The frequent falling over onto their backs prevented us from performing classical assays of motor function such as rotarod or foot printing. Instead, we assayed the righting reflex, open field exploration, and USVs. We also tested for clinically relevant features such as tremor and dystonia-like postures, which often arise with cerebellar dysfunction. We found that $En1^{Cre/+};Atoh1^{fl/-}$ mice perform poorly compared to

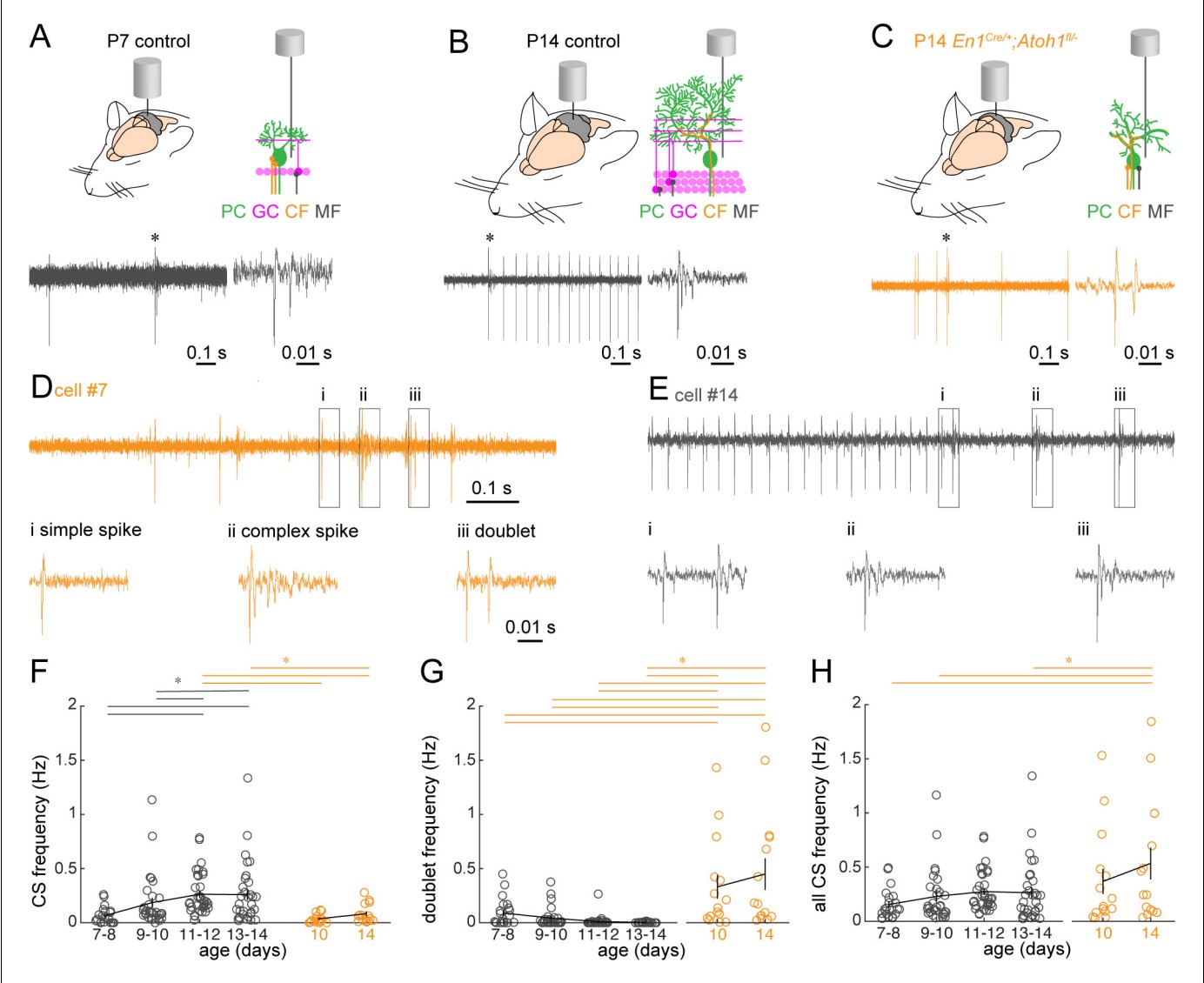

**Figure 8.** Purkinje cell complex spike firing patterns change from P7 to P14 in control mice and are altered in P14 *En1^{Cre/+};Atoh1^{fl/-}* mice. (**A**) Schematic of in vivo Purkinje cell recordings in P7 control mouse. Top left: cerebellum with recording electrode. Top right: schematic showing the recording electrode extracellular and adjacent to the Purkinje cell body. Bottom right: representative 1 s recording from Purkinje cell, each large vertical line is an action potential. *indicates a doublet/complex spike. Bottom left: doublet indicated on the right. (**B**) Same as (**A**) For P14 control mouse. * = a complex spike. (**C**) Same as (**A**) for P14 *En1^{Cre/+};Atoh1^{fl/-}* mouse, * is a doublet. (**D**) Example of a Purkinje cell recording with a classical complex spike and doublet occurring in the same cell (trace is 1 s long). Bottom are expanded views of (i) simple spike, (ii) complex spike, and (iii) doublet in top trace. The trace is an example from a Purkinje cell recording in a P14 *En1^{Cre/+};Atoh1^{fl/-}* mouse. (**E**) Example of a Purkinje cell recording with a train of classical complex spikes (trace is 1 s long). Bottom are expanded views for the three complex spikes occurring in rapid succession of one another (i, ii, iii). (**F**) Frequency of 'classical' complex spikes. (**G**) Frequency of doublets. (**H**) Frequency of combined complex spikes and doublets. For F-H, significance was determined using an ANOVA between the four control age groups and two *En1^{Cre/+};Atoh1^{fl/-}* age groups followed by a Tukey-Kramer post-hoc analysis to assess differences between the groups. Significance was accepted at $p < 0.05$. In the figure, the statistical significance between two groups is indicated as a line starting and ending above the two groups. Gray lines indicate differences within control groups of different ages and orange lines indicate differences between control and *En1^{Cre/+};Atoh1^{fl/-}* groups. N/n-numbers: control P7-8: n=22 cells from N=12 mice; control P9-10: n=30/N=12; control P11-12: n=35/N=11; control P13-14: n=32/N=14; *En1^{Cre/+};Atoh1^{fl/-}* P10: n=15/N=3; *En1^{Cre/+};Atoh1^{fl/-}* P14: n=15/N=6. The raw data and specific p-values for all the comparisons are presented in *Figure 8—source data 1*.

The online version of this article includes the following source data for figure 8:

**Source data 1.** Source data and specific p-values for representative graphs in *Figure 8*.

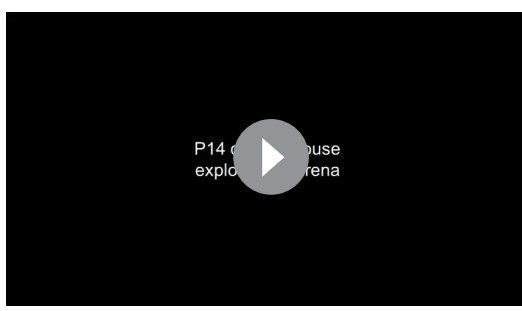

**Video 1.** $En1^{Cre/+};Atoh1^{fl/-}$ mice have a spectrum of visible motor impairments. At P14, control mice explore the open box smoothly, whereas $En1^{Cre/+};Atoh1^{fl/-}$ mice have a visible tremor, often fall on their backs, and have dystonia-like postures and movements in their hindlimbs.

https://elifesciences.org/articles/68045#video1

control littermates during the righting reflex. They were significantly slower in returning onto four paws when compared to $Atoh1^{fl/+}$ control mice at P8 and P10, and they were slower than heterozygous $En1^{Cre/+};Atoh1^{fl/+}$ mice only at P10 (*Figure 9B*). Because all mice attempted to turn right-side-up immediately after being placed on their backs, it is likely that this delay in righting is the result of impaired motor coordination rather than an abnormal sense of gravity. Next, we tested whether $En1^{Cre/+};Atoh1^{fl/-}$ mice showed abnormal USVs when briefly separated from their mothers (*Figure 9C*). We found that call time in $En1^{Cre/+};Atoh1^{fl/-}$ mice was shorter than those observed in control littermates and that $En1^{Cre/+};Atoh1^{fl/-}$ mice called less frequently than $Atoh1^{fl/+}$ mice (*Figure 9C–E*). We next quantified how $En1^{Cre/+};Atoh1^{fl/-}$ mice moved in an open field (*Figure 9F*). The distance traveled or movement time in a 15-min-period was not significantly impaired (total distance (cm): $Atoh1^{fl/+}$: 58.1±11.7; $En1^{Cre/+};Atoh1^{fl/+}$: 35.7±9.9; $Atoh1^{fl/-}$: 44.4±6.5; $En1^{Cre/+};Atoh1^{fl/-}$: 43.1±12.4; Kruskal-Wallis test p=0.32; movement time (s): $Atoh1^{fl/+}$: 55.9±5.9; $En1^{Cre/+};Atoh1^{fl/+}$: 42.1±7.2; $Atoh1^{fl/-}$: 50.1±7.4; $En1^{Cre/+};Atoh1^{fl/-}$: 83.5±17.7; Kruskal-Wallis test p=0.14). However, $En1^{Cre/+};Atoh1^{fl/-}$ mutant mice traveled slower than $Atoh1^{fl/-}$ control mice and the $En1^{Cre/+};Atoh1^{fl/+}$ mice made more isolated movements during their trajectory compared to all their littermate controls (*Figure 9F–H*). Finally, we observed a tremor in the mutants and measured the severity with our custom-made tremor monitor (*Figure 9I*; *Brown et al., 2020*). We found that $En1^{Cre/+};Atoh1^{fl/-}$ mice had a higher power tremor in the 12–16 Hz frequency range (*Figure 9J*). This range corresponds to physiological tremor and indicates the presence of a pathophysiological defect that emerges from a rise in baseline values. The mutant mice also had a higher peak tremor power compared to all control littermates (*Figure 9K*). Together, we uncovered that the lack of rRL-derived excitatory cerebellar neurons in developing $En1^{Cre/+};Atoh1^{fl/-}$ mice leads to multiple abnormal cerebellar-dependent behaviors that are observed in the first 2 weeks of postnatal life.

## Discussion

In this paper, we used $En1^{Cre/+};Atoh1^{fl/-}$ mice as a model of cerebellar agranularity to test how cell-to-cell interactions impact the formation of functional circuits. Using this model with circuit-wide loss of granule cell neurogenesis, we uncovered how these late-born cells influence the functional development of their downstream synaptic partners, the Purkinje cells. We find that granule cell elimination stalls the anatomical and functional maturation of postnatal Purkinje cells. In humans, granule cell neurogenesis is impaired in premature infants with cerebellar hemorrhages, which is expected as proliferating granule cell precursors are highly vulnerable to the hemorrhage, likely because of their high metabolic demand (*Dobbing, 1974*; *Gano and Barkovich, 2019*; *Hortensius et al., 2018*). Moreover, even a localized lesion that occurs during early cerebellar genesis can have major repercussions; loss of just a few precursors has exponential effects on the number of granule cells that integrate into the cerebellar circuit (*Corrales et al., 2004*; *Corrales et al., 2006*). Our findings show that manipulating granule cell neurogenesis not only causes major anatomical defects, but it also obstructs the remodeling and wiring of circuits that establish cerebellar cortical function during a dynamic postnatal period. Our data add to our understanding of how temporally defined insults to the developing cerebellum may result in downstream circuit dysfunctions that are associated with specific neurological disorders observed in premature infants (*Dijkshoorn et al., 2020*).

In our model of agranularity, we show that deletion of *Atoh1* from the *En1* domain impairs the development of the majority of granule cells. Even though some granule cells escape our genetic manipulation, we are confident that we impair the neurogenesis of the majority of granule cells (*Figure 1—figure supplement 3*). Granule cells are the major driver for increasing cerebellar size,

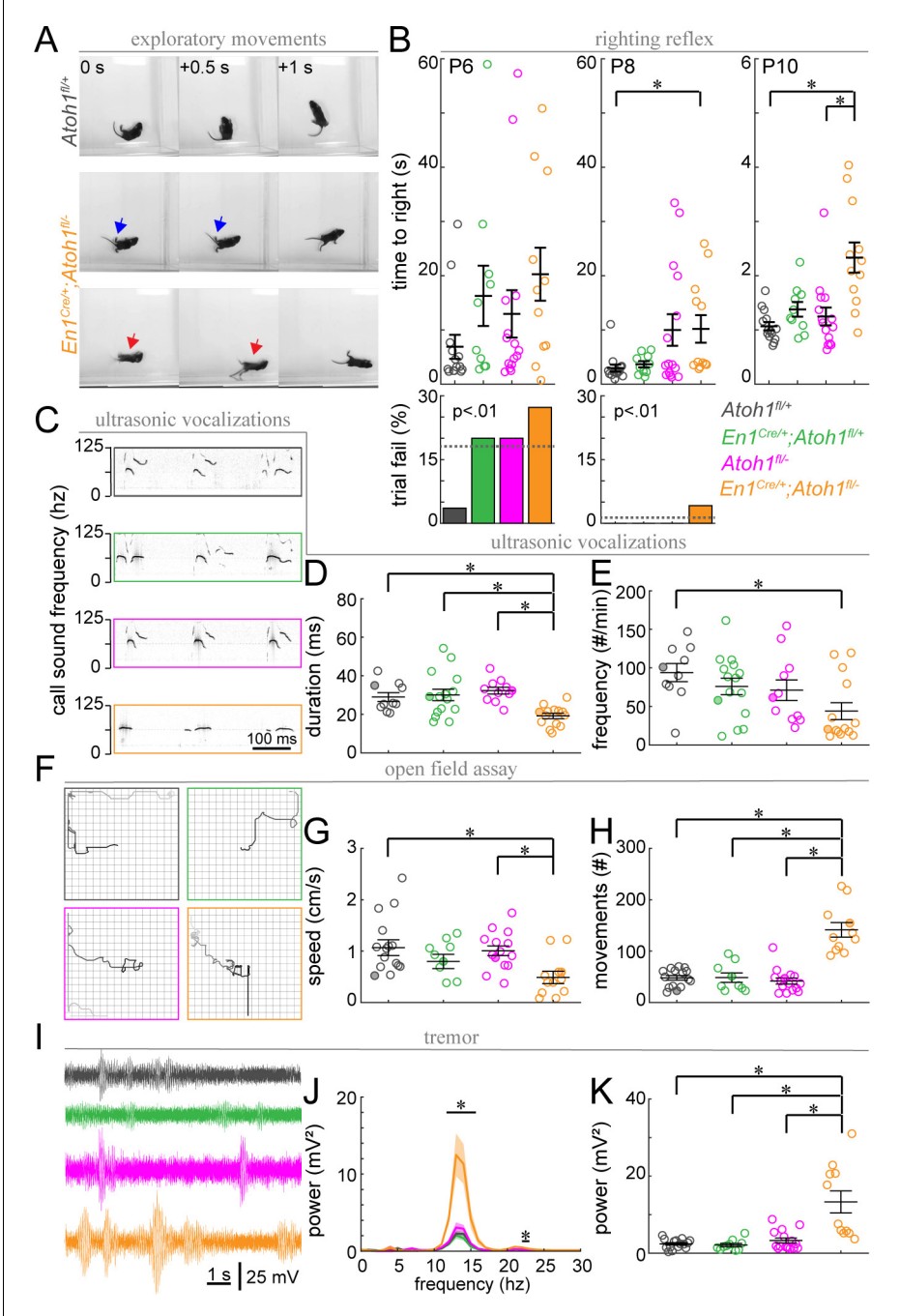

**Figure 9.** Postnatal *En1^{Cre/+};Atoh1^{fl/-}* mice have abnormal motor coordination, enhanced tremor and altered vocalization behavior. (**A**) Timed-series photos of *Atoh1^{fl/+}* (gray) and *En1^{Cre/+};Atoh1^{fl/-}* (orange) mice. *En1^{Cre/+}; Atoh1^{fl/-}* mice have a wide stance (blue arrows) and fall on their backs (red arrows). (**B**) Time to right in the righting reflex of P6, P8, and P10 mice (top) and percentage of failed trials (bottom). (**C**) Representative ultrasonic vocalization traces with intensity of the black line representing the power of the vocalization frequency (filled circles in **D** and **E**). (**D**) Duration of vocalizations. (**E**) Frequency of vocalizations. (**F**) Representative tracks of mice in the open field (filled circles in **G** and **H**). Box measures 40x40 cm. (**G**) Movement speed. (**H**) Number of movements. (**I**) Representative power spectra of tremor recordings. (**J**) Tremor power at different frequencies. (**K**) Peak tremor power. N-numbers: *Atoh1^{fl/+}* (gray): N=10–15 mice; *En1^{Cre/+};Atoh1^{fl/+}* (green): N=9–15 mice; *Atoh1^{fl/-}* (pink): N=11–15 mice; *En1^{Cre/+};Atoh1^{fl/-}* (orange): N=11–14 mice. Significance was determined using a non-parametric Kruskal-Wallis test followed by a Tukey-Kramer post-hoc test. *p<0.05. The raw data and specific p-values for all the comparisons are presented in *Figure 9—source data 1*.

*Figure 9 continued on next page*

Figure 9 continued

The online version of this article includes the following source data for figure 9:

**Source data 1.** Source data and specific p-values for representative graphs in *Figure 9*.

foliation, and lamination. Accordingly, the cerebella of $En1^{Cre/+};Atoh1^{fl/-}$ mice do not undergo proper expansion, and are devoid of lobules and a densely packed granule cell layer (*Figure 1D–J*). In addition, the granule cell marker, GABAR$\alpha$6, is not reliably expressed in $En1^{Cre/+};Atoh1^{fl/-}$ mice (*Figure 1—figure supplement 2A*) and we find an approximate thousand-fold reduction in total *En1;Atoh1* lineage neurons (*Figure 1—figure supplement 3*). Furthermore, the density of Vglut1$^+$ terminals is decreased nearly nine-fold, with the remaining Vglut1 signal mainly representing mossy fiber terminals since any granule cell and unipolar brush cell escapees would only contribute minimally to the existing circuit in the mutants (*Figure 4*). Regardless, even if some granule cells are present, they are not sufficient to drive the cellular circuit reorganizations that are known to depend on granule cells (*Figure 2*, *Figure 7*).

In addition to impaired granule cell neurogenesis, we observe a lower number of unipolar brush cells and excitatory nuclei cells in our $En1^{Cre/+};Atoh1^{fl/-}$ mice, which could, in theory, contribute to our findings of abnormal Purkinje cell function by causing a lack of feedforward and feedback signals, respectively (*van Dorp and De Zeeuw, 2015*; *Gao et al., 2016*). However, both excitatory nuclei cells and unipolar brush cells signal through the granule cells, so loss of granule cells alone would in theory have a similar effect on Purkinje cell function as the combined loss of all three subtypes. In addition, a lower number of excitatory nuclei cells can reduce the number of Purkinje cells (*Willett et al., 2019*) and a lower number of Purkinje cells was also noted in models of chimeric *Atoh1* knockouts (*Jensen et al., 2004*); therefore, the reduction in cerebellar size that we reported in *Figure 1* could be, at least in part, enhanced by a lower number of Purkinje cells. Regardless, the $En1^{Cre/+};Atoh1^{fl/-}$ cerebellum is filled with Purkinje cells, the predominant neuronal type comprising this massively altered cerebellum, making it an ideal environment for identifying and measuring synapse densities and easily isolating single-units during the in vivo recordings.

Nevertheless, there are several caveats to using an agranular model to investigate circuitry. For instance, loss of morphogenetic processes that determine cerebellar architecture including its size (*Dahmane and Ruiz i Altaba, 1999*), foliation (*Corrales et al., 2006*), and layering (*Miyata et al., 2010*) complicate interpretations of how Purkinje cells directly respond to granule cells. A previous study showed that impaired granule cell migration can also impair Purkinje cell morphology and monolayer formation, which in turn could change Purkinje cell function (*Adams et al., 2002*). However, here, we observed that the firing patterns of young Purkinje cells (P7-P10) do not differ between control and experimental mice, even though the defects in monolayer formation and foliation have already occurred by this time. This suggests that our observations of abnormal maturation in Purkinje cell firing patterns cannot be solely attributed to gross abnormalities in the anatomical structure of the cerebellum during morphogenesis in the $En1^{Cre/+};Atoh1^{fl/-}$ mice. In addition, changes in cerebellar shape and size could influence the function of forebrain regions (*Kuemerle et al., 2007*), and the vast connectivity of the cerebellum with forebrain regions such as the hippocampus and prefrontal cortex could contribute to the non-motor vocalization defects that we observed (*Liu et al., 2020*; *McAfee et al., 2019*). The regional specificity of cerebellar circuitry that mediates non-motor connectivity is thus likely obscured in $En1^{Cre/+};Atoh1^{fl/-}$ mice (*Badura et al., 2018*; *Stoodley and Limperopoulos, 2016*).

Despite these caveats, much can be learned from our genetically precise agranular mouse model. Specifically, our mouse model has several advantages over previously described agranular mice when it comes to investigating the contribution of granule cell neurogenesis to cerebellar development and function. Because our model takes advantage of a conditional genetic strategy that only targets the rhombic lip lineage, our manipulation does not affect cell intrinsic developmental Purkinje cell programs (*Herrup, 1983*; *Miyata et al., 2010*; *Sheldon et al., 1997*). And unlike previous models, our approach is independent of procedural variations (*Altman and Anderson, 1971*; *Sathyanesan et al., 2018*; *Yoo et al., 2014*), targets the entire *Atoh1* lineage in the cerebellum (*Ben-Arie et al., 1997*; *Jensen et al., 2002*; *Jensen et al., 2004*), and has allowed us to study postnatal development. As a result, mutant mice with the $En1^{Cre/+};Atoh1^{fl/-}$ genotype have highly

penetrant, consistent, and reliable anatomical and functional phenotypes, all of which have provided us with key insights into how cerebellar lineages shape circuit development and behavior.

The cerebellum controls motor and non-motor behaviors (*Hull, 2020*; *Wagner and Luo, 2020*). Regardless of the specific behavior, Purkinje cells are always at the center of the responsible circuit. Interestingly, during the first 2 weeks of life in mice, motor control becomes more precise (*Lalonde and Strazielle, 2015*), in concert with the refinement of the cerebellar circuitry (*White and Sillitoe, 2013*). During this period, Purkinje cell innervation switches from climbing fibers and mossy fibers to climbing fibers and parallel fibers (*Mason and Gregory, 1984*), climbing fibers are pruned and a 'winner' establishes a single Purkinje cell target (*Kano et al., 2018*), and Purkinje cell zones are sharpened (*White et al., 2014*). Accompanying these changes are emergent properties of the two Purkinje cell spike profiles, simple spikes and complex spikes. In addition to the increased complexity of intrinsic cellular properties that have been defined using slice electrophysiology approaches (*McKay and Turner, 2005*), we postulated using our in vivo recordings that intercellular interactions between granule cells and Purkinje cells during development may support the maturation of Purkinje cell firing properties, both by providing direct inputs and by supporting anatomical maturation, including the outgrowth of Purkinje cell dendrites. In control mice, we observed dynamic changes in normal Purkinje cell firing between P7 and P14. We found an increase in firing rate that was not uniformly acquired but was present during bursts of rapid firing that were interspersed with frequent pauses without Purkinje cell action potentials. We previously reported burst-like Purkinje cell firing from P15 to P19, although by P30 the pattern acquires the regularity that is characteristic of adults (*Arancillo et al., 2015*). Thus, burst-like firing occurs at intermediate stages of normal Purkinje cell development. Interestingly, bursting Purkinje cell firing patterns are also observed in mouse models of ataxia, tremor, and dystonia (*Brown et al., 2020*; *Fremont et al., 2014*; *LeDoux and Lorden, 2002*; *Miterko et al., 2019*; *Miterko et al., 2021*; *White and Sillitoe, 2017*; *White et al., 2016*). The dynamically adapting circuit in control mice and the range of disease severities in disease models with bursting Purkinje cells raise the possibility that Purkinje cell firing is differentially decoded by downstream neurons based on the age of the mice. The data also indicate that the intermediate stages of Purkinje cell development not only highlight a developmental phase characterized by erratic neuronal activity, but that this mode of firing represents a pathophysiological hallmark that could be a default network state in different diseases. The higher level of baseline tremor in $En1^{Cre/+};Atoh1^{fl/-}$ mice suggests the possibility that perhaps the process of stabilizing neuronal activity is affected in our mutants. In contrast, alterations in other behaviors such as the USVs indicates that behaviors required during the postnatal period, such as communication with the dams, are also dependent of stage-specific neuronal properties. Therefore, the presence of granule cells in the cerebellar cortex impacts behaviors used at different ages as a consequence of how they influence Purkinje cell firing.

The behavioral abnormalities in $En1^{Cre/+};Atoh1^{fl/-}$ mice are likely caused by a combined effect of Purkinje cell dysfunction and a reduction of excitatory neurons in the cerebellar nuclei. Given the broad network effects of our manipulation, it is not surprising that $En1^{Cre/+};Atoh1^{fl/-}$ mice have such severe behavioral abnormalities. However, our findings are very interesting given what we know about cerebellar function. First, while $En1^{Cre/+};Atoh1^{fl/-}$ mice often rotate onto their backs and have a slower righting reflex (*Figure 9A–B*, *Video 1*), they immediately initiate righting in both cases, suggesting that these mice have normal perception of gravitational force and that the observed phenotypes are mainly due to motor impairments. Second, because some extra-cerebellar structures are involved in airway pressure and breathing (*Bautista and Dutschmann, 2014*; *Chamberlin, 2004*; *Dutschmann and Herbert, 2006*), specifically the parabrachial nucleus and Kölliker Fuse (*van der Heijden and Zoghbi, 2018*), the defects in USVs could ultimately be due to alterations in multiple brain regions with particular effects on the duration of vocalizations (*Figure 9C–D*). Yet, the respiratory cell types we have manipulated are unlikely to drive the initiation and number of vocalizations (*Figure 9E*), suggesting that these abnormalities are cerebellum-driven. Finally, loss of Purkinje cell signaling is sufficient to reduce baseline and harmaline-induced tremor (*Brown et al., 2020*), whereas we observe that loss of excitatory cerebellar neurons causes an enhanced tremor phenotype (*Figure 9I–K*). These data show that unlike Purkinje cells, excitatory cerebellar neurons are not necessary for propagating tremor but they are nevertheless required for modulating it to achieve adequate control of the muscles.

The severity of motor impairments and electrophysiological changes in Purkinje cells in our model and previously reported agranular mice remains in contrast to the modest changes in Purkinje cell firing after silencing granule cell synapses (*Galliano et al., 2013*). However, a noteworthy distinction should be made that in our mice, granule cells are removed altogether, whereas *Galliano et al., 2013* blocked granule cell neurotransmitter release – the difference in functional outcomes could therefore include unique cell non-autonomous impacts on development induced by each type of manipulation and not solely depend on the primary manipulated cell type, the granule cells. Furthermore, in multiple models, impairing parallel fiber synapses results in motor impairments that can be assessed with the rotor rod assay (*Aiba et al., 1994*; *Park et al., 2019*), which we could not do due to the severity of motor impairment in $En1^{Cre/+};Atoh1^{fl/-}$ mice. Taking these results together, from both the technical and conceptual standpoints, when one seeks to resolve developmental mechanisms, we must not only consider what is manipulated, but also how it is manipulated. As such, the timing of neurogenesis is a primary consideration. Purkinje cells are generated between E10 and E13 (*Hashimoto and Mikoshiba, 2003*) and granule cell progenitors from ~E13 onwards (*Machold and Fishell, 2005*; *Rose et al., 2009*; *Wang et al., 2005*). Whereas Purkinje cells migrate into the core of the cerebellar anlage upon their birth, granule cell precursors first migrate over the surface of the developing cerebellum and proliferate extensively in the external granular layer to increase the precursor pool (*Wingate and Hatten, 1999*). Only after this phase do they migrate radially past Purkinje cells, the first potential opportunity for direct cell-to-cell interactions. Based on our data, we argue that the initial communication between Purkinje cells and granule cells sets the efficiency of Purkinje cell function and the establishment of Purkinje cell spike properties through structural as well as synaptic signals. Thus, insults to granule cell proliferation and an obstruction of granule cell neurogenesis may have different, and perhaps more severe effects on downstream Purkinje cell function, compared to lesions of mature granule cells. This may explain why relatively mild but early lesions and structural changes, as observed in preterm infants with and without cerebellar hemorrhage, have large effects on cerebellar function.

# Materials and methods

## Key resources table

| Reagent type (species) or resource | Designation | Source or reference | Identifiers | Additional information |
|---|---|---|---|---|
| Antibody | Anti-Calbindin (guinea-pig polyclonal) | Synaptic Systems | RRID:AB_2619902 | IF: (1:1,000) |
| Antibody | Anti-GABARα6 (rabbit polyclonal) | Millipore Sigma | RRID:AB_91935 | IF: (1:500) |
| Antibody | Anti-Tbr2 (rabbit polyclonal) | Abcam | RRID:AB_778267 | IF: (1:500) |
| Antibody | Anti-Calretinin (mouse monoclonal) | Swant | RRID:AB_10000320 | IF: (1:500) |
| Antibody | Anti-NFH (mouse monoclonal) | Biolegend | RRID:AB_2564642 | IF: (1:1,000) |
| Antibody | Anti-HCN1 (rabbit polyclonal) | Alomone Lab | RRID:AB_2039900 | IF: (1:500) |
| Antibody | Anti-RORα (goat polyclonal) | Santa Cruz | RRID:AB_2301066 | IF: (1:250) |
| Antibody | Anti-PV (rabbit polyclonal) | Swant | RRID:AB_10000344 | IF: (1:1,000) |
| Antibody | Anti-neurogranin (rabbit polyclonal) | Millipore | RRID:AB_91937 | IF: (1:500) |
| Antibody | Anti-ZebrinII (mouse monoclonal) | Gift from Dr. Hawkes | | IF: (1:500) |
| Antibody | Anti-Vglut1 (rabbit polyclonal) | Synaptic Systems | RRID:AB_887877 | IF: (1:500) |
| Antibody | Anti-Vglut2 (rabbit polyclonal) | Synaptic Systems | RRID:AB_887883 | IF: (1:500) |

*Continued on next page*

*Continued*

| Reagent type (species) or resource | Designation | Source or reference | Identifiers | Additional information |
|---|---|---|---|---|
| Antibody | Anti-PLCβ4 (rabbit polyclonal) | Santa Cruz Biotechnology | RRID:AB_654082 | IF: (1:150) |
| Chemical compound, drug | WGA-Alexa 555 | Thermo Fisher Scientific | | |
| Chemical compound, drug | Fast green | Sigma-Aldrich | | |
| Chemical compound, drug | PBS | Sigma-Aldrich | | |
| Commercial assay or kit | Golgi-Cox staining kit | FD Neurotechnologies | | |
| Strain, strain background (*M. musculus*) | Mouse: Atoh1-FlpO | Gift from Dr. Zoghbi | | *van der Heijden and Zoghbi, 2018* |
| Strain, strain background (*M. musculus*) | Mouse: Rosa-FSF-LSL-tdTomato (Ai65(RCFL-tdT), Gt(ROSA)26Sortm65.1 (CAG-tdTomato)Hze) | The Jackson Laboratory | RRID:IMSR_JAX:021875 | |
| Strain, strain background (*M. musculus*) | Mouse: En1-Cre (En1tm2 (cre)Wrst/J) | The Jackson Laboratory | RRID:IMSR_JAX:007916 | |
| Strain, strain background (*M. musculus*) | Mouse: Atoh1-Flox (Atoh1tm3Hzo) | The Jackson Laboratory | RRID:MGI:4420944 | |
| Software, algorithm | MATLAB | MathWorks | RRID:SCR_001622 | |
| Software, algorithm | ImageJ | National Institutes of Health | RRID:SCR_003070 | |
| Software, algorithm | Photoshop | Adobe | RRID:SCR_014199 | |
| Software, algorithm | Illustrator | Adobe | RRID:SCR_010279 | |
| Software, algorithm | Spike2 | CED | RRID:SCR_010279 | |
| Other | Tungsten electrodes | Thomas Recording | | |
| Other | VetBond | 3M | | |

## Animals

All mice used in this study were housed in a Level 3, AALAS-certified facility. All experiments and studies that involved mice were reviewed and approved by the Institutional Animal Care and Use Committee (IACUC) of Baylor College of Medicine (BCM). The following transgenic mouse lines were used for the experiments carried out in this study: $Atoh1^{FlpO}$ (*van der Heijden and Zoghbi, 2018*); $En1^{Cre}$ ($En1^{tm2(cre)Wrst/J}$, JAX:007916); $Ai65$ ($Gt(ROSA)26Sor^{tm65.1(CAG-tdTomato)Hze}$, JAX:021875); $Atoh1^{Flox}$ ($Atoh1^{tm3Hzo}$, MGI:4420944). Conditional knockout mice were generated by crossing $En1^{Cre/+};Atoh1^{+/FlpO}$ double heterozygote mice (which can be functionally defined as $En1^{Cre/+};Atoh1^{+/-}$ mice) with homozygote $Atoh1^{fl/fl}$ mice. $En1^{Cre/+};Atoh1^{fl/-}$ mice were considered experimental conditional knockout mice and all littermates with a different genotype were considered control mice. Specifically, we used $Atoh1^{fl/+}$, $Atoh1^{FlpO/fl}$, and $En1^{Cre/+};Atoh1^{fl/+}$ mice as controls. Ear tissue or tail clips were collected before weaning and used for genotyping and identification of the different alleles. For all experiments, we bred mice using standard timed pregnancies, noon on the day a vaginal plug was detected was considered embryonic day E0.5 and P0 was defined as the day of birth. Pups of both sexes were used in all experiments.

## Tissue processing

Brain tissue was collected as described in our previous publications (*Zhou et al., 2020*). First, we anesthetized mice with Avertin. Once the mice did not respond to toe or tail pinch, we accessed the chest cavity and then penetrated the heart with a butterfly needle for perfusions. The mice were perfused with 1M phosphate-buffered saline (PBS pH 7.4) to remove blood from the tissue and 4% paraformaldehyde (PFA) to fix the tissue. The tissue was concomitantly post-fixed overnight in 4% PFA at 4°C. Tissue was cryoprotected in a sucrose gradient (10% → 20% → 30% sucrose in PBS) at 4°C, each step lasting until the tissue sank to the bottom of a 15 mL tube. Tissue was frozen in optimal cutting temperature (OCT) solution and stored at −80°C until it was cut. All tissue were cut into

40 μm free-floating tissue sections and stored in PBS at 4°C until it was used for immunohistochemistry.

## Immunohistochemistry

Free floating tissue sections were stained according to the following protocol. Free floating tissue sections were blocked in 10% normal goat or donkey serum and 0.1% Triton-X in PBS (PBS-T) for two hours. Next, tissue sections were incubated overnight in primary antibodies in blocking solution. Tissue was washed three times for 5 min in PBS-T. For fluorescent staining, the tissue was incubated for 2 hr in PBS-T with the preferred secondary antibodies conjugated to an Alexa fluorophore. Finally, tissue sections were washed three times in PBS-T and mounted on electrostatically coated slides with hard-set, DAPI-containing mounting medium. Alternatively, for DAB staining, we incubated the tissue for 2 hr in PBS-T with the preferred secondary antibodies that was conjugated to horseradish peroxidase (HRP). After washing three times in PBS-T, the tissue was incubated with diaminobenzidine (DAB) solution until the desired color intensity was reached. The DAB color reaction was stopped by washing tissue three times with PBS-T. The tissue was then mounted on electrostatically coated glass slides, dehydrated in an ethanol series (70% → 90% → 100%) and then mounted using Xylene or histoclear. All steps of immunohistochemistry were performed at room temperature. All mounted slides were stored at 4°C until they were imaged.

The following primary antibodies were used for the data described in this manuscript: guinea pig (gp)-α-Calbindin (1:1,000; SySy; #214004); rabbit (rb)-α-gamma-aminobutyric acid receptor α6 (GABARα6; 1:500; Millipore Sigma; #AB5610), rb-α-T-box brain protein 2 (Tbr2; 1:500; Abcam; #AB23345), mouse (ms)-α-Calretinin (1:500; Swant; #6B3); ms-α-Neurofilament Heavy (NFH; 1:1,000; Biolegend; #801701); rb-α-Hyperpolarization Activated Cyclic Nucleotide Gated Potassium Channel 1 (HCN1; 1:500; Alomone Lab; #APC-056); goat (gt)-α-RAR-related orphan receptor alpha (RORα; 1:250; Santa Cruz; #F2510); rb-α-parvalbumin (PV; 1:1,000; Swant; #PV25); rb-α-neurogranin (1:500; Millipore Sigma; #AB5620); ms-α-ZebrinII (1:500; kind gift from Dr. Richard Hawkes, University of Calgary, Calgary, Alberta, Canada); rb-α- PLCβ4 (1:150; Santa Cruz Biotechnology; catalog #sc-20760); rb-α-Vglut1 (1:500; SySy; #135302); rb-α-Vglut2 (1:500; SySy; #135403). The following secondary antibodies were used for immunohistochemistry: HRP-conjugated goat (gt)-α-mouse; gt-α-rabbit; and donkey (dk)-α-goat (1:200; DAKO). The following secondary antibodies were used for immunofluorescence: dk-α-mouse IgG Alexa Fluor 488 (1;1,500; Thermo Fisher Scientific; #A21202) and gt-α-gp IgG Alexa Fluor 488 (1;1,500; Thermo Fisher Scientific;#A11073).

## Cresyl violet staining

Brain tissue sections were mounted on glass slides and then dried overnight. Slides were submerged in 100% histoclear and rehydrated in an ethanol series (100% → 90% → 70%). Then, the slides were submerged in cresyl violet solution for staining until sufficiently dark and then dehydrated in an ethanol series (70% → 90% → 100%). Finally, the slides were sealed with a coverslip using Cytoseal mounting media. All steps were performed at room temperature and the mounted slides were stored at 4°C until they were imaged.

## Neuroanatomical anterograde tracing

Anterograde neuroanatomical tracing of mossy fibers to the cerebellum was performed as described previously (*Lackey and Sillitoe, 2020*; *Sillitoe, 2016*). P12 pups were anesthetized with isoflurane on a surgery rig. Hair was removed and an incision was made in the skin over the lower thoracic/upper lumbar spinal cord, using the curvature of the spine as a guide. We used a Nanoject II to inject 0.2–1 μl of 2% WGA-Alexa Fluor 555 (Thermo Fisher Scientific; #W32464) and 0.5% Fast Green (Sigma-Aldrich; #F7252, used for visualization) diluted in 0.1 M phosphate-buffered saline (PBS; Sigma-Aldric; #P4417; pH 7.4). Tracer was injected 1 mm below the surface of the spinal cord, on the right side of the dorsal spinal vein. After tracer injection, we applied antibiotic ointment and closed the incision using VetBond (3M; #1469 SB) and wound clips (Fine Science Tools; #12032–07). Pups were placed back with the mom after waking up from anesthesia. We placed soft food and hydrogel on the cage floor and monitored behavior closely for whether the mom accepted the pups back into the litter. Tissue was collected (see section on *Tissue processing* above) for tracer visualization 2 days after the surgery, at P14.

## Golgi-Cox staining

Golgi-Cox staining was performed according to previously described protocols (*Brown et al., 2019*) and the manufacturer's instructions (FD Neurotechnologies; #PK401). Brains were dissected from the skulls and immediately emerged in the staining solution. After staining, tissue sections were cut at a thickness of 10 µm and directly mounted onto electrostatically coated glass slides. The tissue was then dehydrated in an ethanol series (70% → 90% → 100%), cleared with Xylene, and mounted with cytoseal. All slides were dried overnight before imaging and were kept at 4°C for storage.

## Microscopy and image processing

Photomicrographs of stained whole mount cerebella and DAB stained cerebellar tissue sections were acquired using Leica cameras DPC365FX and DMC2900, respectively, attached to a Leica DM4000 B LED microscope. Photomicrographs of Golgi-Cox stained tissue sections and WGA-Alexa 555 tracing were captured using Zeiss cameras AxioCam MRc5 and AxiaCam Mrm, respectively, attached to a Zeiss Axio Imager.M2 microscope. Whole mount images were stitched together using Adobe Photoshop (Adobe Systems 2020) Photomerge function. Color brightness and contrast were adjusted using ImageJ (*Schneider et al., 2012*). Photomicrographs of images were cropped to the desired size using Adobe Illustrator (Adobe Systems 2020).

## Image quantification – TdTomato signal density

We obtained high-magnification images from lobule IX in control animals or the ventrocaudal region of conditional knockout mice. We quantified the percentage of the area of interest occupied by the signal by transforming the signal into a binary signal with TdTomato pixel brightness cut off at 80/255.

## Image quantification – unipolar brush cell and excitatory nuclei cell count

We serially sectioned brains and kept every sixth (coronal) or eight (sagittal) section in an individual well of a 24-well plate. This allowed us to have an unbiased tissue collection system that spanned the entire cerebellum. We then stained all tissue sections collected in a single well for anti-Tbr2 (unipolar brush cells) or anti-NFH (excitatory nuclei neurons and Purkinje cells) plus anti-Calbindin (Purkinje cells). Co-staining for NFH and Calbindin was necessary to distinguish between Purkinje cells (Calbindin$^+$ and NFH$^+$) and excitatory nuclei cells (Calbindin$^-$ and NFH$^+$) in primarily the conditional knockout mice, as nuclei neurons and Purkinje cells did not have clear anatomical segregation in these mice. After staining our tissue sections as described above, we imaged whole tissue sections at a magnification that allowed us to identify and count single cells. Images were merged together in photoshop and all Tbr2$^+$ cells and NFH$^+$/Calbindin$^-$ cells were counted in all tissue sections from a single animal in ImageJ. The final cell-count was multiplied by six (coronal) or eight (sagittal) to obtain an estimate of the total number of unipolar brush cells or excitatory nuclei cells for each animal. A t-test was performed to investigate whether there was a statistically significant difference between cell number in control and $En1^{Cre/+};Atoh1^{fl/-}$ mice.

## Image quantification – Purkinje cell morphology analysis

Brightfield images of Golgi-Cox stains were imported into ImageJ and then converted into a binary image with brightness and contrast adjusted so that cellular processes were clearly visible. Then, Sholl analysis was performed using the build-in Sholl analysis module in ImageJ (*Ferreira et al., 2014*) and false positive intersections were manually subtracted from the counts. Next, we used a Zeiss Laser Scanner confocal Microscope model 710 to take high-magnification Z-stacked images (0.2 µm). We imported the images to imageJ/Fiji and used the Simple Neurite Tracer plug-in to track the length of a stained dendrite. We manually counted the number of synaptic boutons along the dendrite and reported this number as a density per dendrite length.

## Image quantification – Vglut1 and Vglut2 density

We stained tissue sections obtained from P7 and P14 control and $En1^{Cre/+};Atoh1^{fl/-}$ mice with anti-Calbindin and anti-Vglut1 or Vglut2 antibodies as described above. Next, we imaged 3–5 tissue sections from three mice per age, per genotype. We imported tissue sections into ImageJ and defined

our area of interest as the area of the image where Vglut1⁺ and Vglut2⁺ synapses could directly innervate Purkinje cells. This included the molecular and Purkinje cell layers in control tissue and the area occupied by Purkinje cell bodies in *En1^{Cre/+};Atoh1^{fl/-}* tissue; both were identified as the area of the image positive for the Calbindin signal. We adjusted the brightness and contrast in the Vglut1 or Vglut2 channels so that individual synapses were visible and then transformed the image into a binary (black and white) image using the Threshold function in ImageJ. We quantified the percentage of area of interest covered by Vglut1 or Vglut2 synapses using the Analyze Particles tool in ImageJ. We also manually counted the number of Purkinje cells that received direct Vglut2⁺ inputs on their somas. Finally, we averaged the values of the area covered by Vglut1⁺ or Vglut2⁺ synapses across all tissue sections (n=3–5) per animal (N=3) and performed a two-way ANOVA (genotype*age) followed by a Tukey-Kramer post-hoc analysis to identify the statistical significance of observed differences in the density of Vglut1⁺ and Vglut2⁺ synapses in mice of different genotypes and ages.

## In vivo electrophysiology

All in vivo, anesthetized experiments were performed as described in previous publications (*Arancillo et al., 2015*; *White and Sillitoe, 2017*). Specifically, we anesthetized mice using a mixture of ketamine 80 (mg/kg) and dexmedetomidine (16 mg/kg). We held mice on a heated surgery pad. We removed hair from skull and made an incision in the skin over the anterior part of the skull. We stabilized the heads of our mice using ear bars and a mouth mount when animals were large enough (most P11-P14 mice) and otherwise fixed the mouse skull (P7-P10 mice) to a plastic mount that was attached to ear bars on our stereotaxic surgery rig to stabilize the head during recordings. Using a sharp needle or dental drill, we made a craniotomy in the interparietal bone plate, ~3 mm dorsal from lambda and ~3 mm lateral from the midline, with a diameter of ~3 mm. We kept our surgical coordinates consistent based on the distance from lambda across mice of all ages, as the skull undergoes significant growth during the ages at which we measured neural activity. After making a craniotomy, we recorded neural activity using tungsten electrodes (Thomas Recording, Germany) and then the digitized the signals into Spike2 (CED, England). We recorded neural activity from cells that were 0–2 mm below the brain surface.

## Analysis of in vivo electrophysiological recordings

All electrophysiological recording data were spike sorted in Spike2. We sorted out three types of spikes: simple spikes, complex spikes, and doublets. Complex spikes were characterized by their large amplitude, and post-spike depolarization and smaller spikelets that follow. Doublets were characterized as action potentials that were followed by one or more smaller action potentials within 20 ms after the initial action potential. All other action potentials were characterized as simple spikes (see examples in *Figures 5*, *6* and *8*). We only included traces with clearly identifiable complex spikes or doublets and analyzed only cells from which we could obtain a sufficiently long and stable recording (186 ± 6.5 s; minimum = 75 s) with an optimal signal to noise ratio.

After spike sorting our traces in Spike 2, we analyzed the frequency and regularity of firing patterns in MATLAB (The MathWorks Inc, version R2018a). For this study, we defined 'frequency' as number of all spikes observed in the total analyzed recording time (spikes/s). We determined the 'frequency mode' by populating the inverse of all observed interspike intervals ($ISI^{-1}$, representing the frequency of a single spike) into 2.5 Hz bins and defining the center of the bin with the largest proportion of spikes as the 'frequency mode' (see *Figure 5D and E* for examples). Pause Percent was the proportion of the recording time during which the ISI was longer than five times the mean ISI for each independent cell, defined as the following: (sum(ISI>5*mean(ISI)))/(total recording time). Our measures of global regularity or burstiness (CV) and regularity (CV2) were based on the interspike intervals (ISI) between two adjacent spikes (in seconds). CV = stdev(ISI)/mean(ISI), and CV2 = mean($2*|ISI_n-ISI_{n-1}|/ (ISI_n+ISI_{n-1})$) (*Holt et al., 1996*).

The rhythmicity index was based on oscillatory properties represented by the auto-correlogram of the ISI (see *Figure 5L and M* for examples) and based on previously described quantifications with minor adjustments (*Lang et al., 1997*; *Sugihara et al., 1995*; *White et al., 2016*). The 'rhythmicity index' was calculated as the sum of the difference in peak and trough of the proportion of spikes at each delay from spike onset, divided by the baseline proportion of spikes. Baseline = (total spike number)²/(recording time/bin width). We used a bin width of 5 ms. The first peak of each

oscillation was determined as the highest bin between 10 ms and 1.5 times the mean ISI for a given cell, and the time of the peak was denoted as $a_1$. Each subsequent peak was determined as the highest bin between the delay-time of the previous trough and $a_n+a_1+10$ ms, where $a_n$ is the time of the previous peak. The first trough was determined as the lowest bin between the first peak ($a_1$) and $a_n+a_1$. Peaks and troughs were only accepted if their sum was higher than four times the standard deviation of auto-correlation between 0.96 and 1 s lag-time, or if the peak was higher than baseline + two times the standard deviation and the trough was lower than baseline – two times the standard deviation.

## Cluster and dimension reduction analysis on simple spikes

To investigate and visualize how firing patterns changed over time, we performed additional computational analysis of the firing properties of single neurons (n=179 neurons). We included the parameters summarized in *Figure 5* in our analysis (frequency, frequency mode, pause percent, rhythmicity index, CV, and CV2). First, we performed an unbiased cluster analysis on the firing properties of single neurons using the MATLAB function 'clusterogram,' which previously allowed us to visualize genotype-specific changes in Purkinje cell firing patterns (*van der Heijden et al., 2021a*). After analysis, we de-identified the cells and labelled them by age and genotype (*Figure 6B*). Next, we performed a t-distributed stochastic neighbor embedding (tSNE) dimension reduction analysis to visualize the relative similarities of firing patterns between cells using the MATLAB function 'tsne' based on Euclidean distance. We plotted tSNE component 1 and 2 against each other (tSNE(1) and tSNE(2)) and color-coded the coordinates based on age and genotype of the cell recording to generate an electrophysiological pseudotimeline (*Figure 6C*). Finally, we used the MATLAB function 'linkage' on the first two components of the principal component analysis on the average firing properties per age and genotype group to define how the firing patterns of each group clustered together. We visualized this unbiased linkage analysis using the MATLAB function 'dendrogram' (*Figure 6D*).

## Behavioral analyses

Righting reflex was measured on P6, P8, and P10 as follows. The mouse was placed on its back in a clean cage without bedding. One finger was used to stabilize the mouse on its back. The timer was set the moment the experimenter removed their finger, and time was recorded until the mouse righted itself up onto its four paws. All mice were tested twice at each age. A 'failed' trial was defined as when the mouse did not right itself within one minute (sixty seconds).

At P7, we recorded pup vocalizations as described previously (*Yin et al., 2018*). Pups were placed in an anechoic, sound-attenuating chamber (Med Associates Inc). The pup was placed in a round plastic tub that was positioned near a CM16 microphone (Avisoft Bioacoustics) that was located in the center of the chamber. Sound was amplified and digitized using UltraSoundGate 416H at a 250 kHz sampling rate and bit depth of 16. Avisoft RECORDER software was used to collect the recordings. Ultrasonic vocalizations were monitored for 2 min for each animal.

We also performed an open-field assay at P13, as previously described (*Alcott et al., 2020*). Mice were habituated to a room with the light set to 200 lux and ambient white noise to 60 dB. We placed each mouse in the center of an open field (40x40x30 cm chamber). The chamber has photobeams that records movement. Each mouse was tested for 15 min and activity was recorded using Fusion software (Accuscan Instruments). We used the standard settings for movement segmentation. We analyzed the data for total distance traveled, movement time, speed, and total movements during the 15-min test period.

We measured tremor using our custom-built tremor monitor (*Brown et al., 2020*). Each mouse was placed in the tremor chamber, which is a translucent box with an open top that is suspended in the air by eight elastic cords that are attached to four metal rods. An accelerometer is attached to the bottom of the box. Mice were allowed to habituate to the chamber for 120 s prior to initiating the tremor recordings. The mice are free to move around in the box. Power spectra of the tremor recordings were assessed using Fast Fourier transform (FFT) with a Hanning window in Spike2 software as previously described (*Brown et al., 2020*). FFT frequency was targeted to ~1 Hz per bin.

## Statistical analyses

Statistical analyses were performed in MATLAB. For the Purkinje cell morphology data, we performed a linear mix-model analysis with genotype as the fixed variable and mouse number as the random variable and accepted $p < 0.05$ as statistically significant. For Vglut1/2 quantification, we averaged measurements from three to five tissue sections per mouse and performed a two-way ANOVA (genotype*age) followed by a Tukey-Kramer post-hoc analysis to define statistical significance between independent groups. For electrophysiology data (frequency (complex spikes and simple spikes), frequency mode, pause percent, rhythmicity index, CV, CV2, and doublet frequency), analysis was performed using an ANOVA with six total groups (four control groups, two mutant groups), followed by a Tukey-Kramer post-hoc test. For behavioral data, we performed a Kruskal-Wallis test followed by a Tukey-Kramer post-hoc test to define significance between independent groups. For these tests, we accepted $p < 0.05$ as statistically significant.

## Acknowledgements

This work was supported by Baylor College of Medicine (BCM), Texas Children's Hospital, The Hamill Foundation, and the National Institutes of Neurological Disorders and Stroke (NINDS) R01NS089664, R01NS100874, and R01NS119301 to RVS. Research reported in this publication was supported by the Eunice Kennedy Shriver National Institute of Child Health and Human Development of the National Institutes of Health under Award Number P50HD103555 for use of the Cell and Tissue Pathogenesis Core and the Mouse Neurobehavioral Core facilities. The content is solely the responsibility of the authors and does not necessarily represent the official views of the National Institutes of Health. Support was also provided by a Dystonia Medical Research Foundation (DMRF) postdoctoral award to MEvdH and an F31NS101891 to AMB. HYZ is supported by the Howard Hughes Medical Institute (HHMI).

## Additional information

### Competing interests

Roy V Sillitoe: Reviewing editor, *eLife*. Huda Y Zoghbi: Senior editor, *eLife*. The other authors declare that no competing interests exist.

### Funding

| Funder | Grant reference number | Author |
| --- | --- | --- |
| Dystonia Medical Research Foundation | | Meike E van der Heijden |
| Howard Hughes Medical Institute | | Huda Y Zoghbi |
| National Institute of Neurological Disorders and Stroke | R01NS089664 | Roy V Sillitoe |
| National Institute of Neurological Disorders and Stroke | R01NS100874 | Roy V Sillitoe |
| National Institute of Neurological Disorders and Stroke | R01NS119301 | Roy V Sillitoe |
| National Institute of Neurological Disorders and Stroke | F31NS101891 | Amanda M Brown |

The funders had no role in study design, data collection and interpretation, or the decision to submit the work for publication.

### Author contributions

Meike E van der Heijden, Conceptualization, Data curation, Software, Formal analysis, Funding acquisition, Validation, Investigation, Visualization, Methodology, Writing - original draft, Project administration, Writing - review and editing; Elizabeth P Lackey, Tao Lin, Data curation,

Investigation, Methodology, Writing - review and editing; Ross Perez, Data curation, Investigation, Visualization, Methodology, Writing - original draft, Writing - review and editing; Fatma S Işleyen, Investigation, Methodology, Writing - review and editing; Amanda M Brown, Formal analysis, Writing - review and editing; Sarah G Donofrio, Writing - review and editing; Huda Y Zoghbi, Conceptualization, Writing - review and editing; Roy V Sillitoe, Conceptualization, Resources, Supervision, Funding acquisition, Project administration, Writing - review and editing

### Author ORCIDs
Meike E van der Heijden ◉ https://orcid.org/0000-0003-0801-8806
Amanda M Brown ◉ http://orcid.org/0000-0002-1484-8972
Sarah G Donofrio ◉ http://orcid.org/0000-0003-1680-3302
Huda Y Zoghbi ◉ http://orcid.org/0000-0002-0700-3349
Roy V Sillitoe ◉ https://orcid.org/0000-0002-6177-6190

### Ethics

Animal experimentation: Animal experimentation: This study was performed in strict accordance with the recommendations in the Guide for the Care and Use of Laboratory Animals of the National Institutes of Health. All animals were housed in an AALAS-certified facility on a 14hr light cycle. Husbandry, housing, euthanasia, and experimental guidelines were reviewed and approved by the Institutional Animal Care and Use Committee (IACUC) of Baylor College of Medicine (protocol number: AN-5996).

### Decision letter and Author response
Decision letter https://doi.org/10.7554/eLife.68045.sa1
Author response https://doi.org/10.7554/eLife.68045.sa2

## Additional files

### Supplementary files
• Transparent reporting form

### Data availability

All data generated or analysed during this study are included in the manuscript and supporting files. Source data files have been provided for Figures 1, 3–9.

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
