## [Decision Letter]

**Acceptance summary:**

Cerebellar granule cells provide extensive excitatory inputs to Purkinje neurons. Using genetic tools to abolish granule cells, this study explores how granule cells could regulate the development of structural, synaptic and intrinsic firing properties of Purkinje neurons. The results from this study contribute to our understanding of cerebellar development and how developmental insults to the cerebellum may lead to motor disorders.

**Decision letter after peer review:**

[Editors’ note: the authors submitted for reconsideration following the decision after peer review. What follows is the decision letter after the first round of review.]

Thank you for submitting your work entitled "Maturation of Purkinje cell firing properties relies on granule cell neurogenesis" for consideration by *eLife*. Your article has been reviewed by 3 peer reviewers, one of whom is a member of our Board of Reviewing Editors, and the evaluation has been overseen by a Senior Editor. The reviewers have opted to remain anonymous.

Our decision has been reached after consultation between the reviewers. Based on these discussions and the individual reviews below, we regret to inform you that your work will not be considered further for publication in *eLife*.

The reviewers appreciate the authors for collecting this data set recording the various effects of loss of granule cells on Purkinje cells and motor behaviors. However, reviewers also pointed out several weaknesses in the manuscript. Primarily, the manuscript feels like a collection of unrelated observations as no clear causal link is established between the various cellular/behavioral phenotypes. Secondly, some of the phenotypes reported have also been seen in other models of agranularity and the results presented are along expected lines, taking away from the novelty of their study. Thirdly, several of the claims need stronger support and quantification with accompanying statistical testing. Please see the individual reviews appended below for more detail on these comments. We grappled with whether we could consider a revision that addressed these concerns. In the event, they are rather substantive and attending to them will take time and experimental effort, This will not be feasible in the time we usually expect. Should the authors be able to address all concerns fully, they may want to consider a fresh submission to *eLife*.

*Reviewer #1:*

In this manuscript, Sillitoe and colleagues use a novel conditional knockout mouse line to ask how the absence of granule cells affects Purkinje neuron development and function. By knocking out the Atoh1 gene in the Engrailed1 lineage, they generate mice that are 'agranular' – i.e. lacking cerebellar granule cells. This mouse has previously been studied by two of the authors with respect to respiratory networks (van der Heijden and Zoghbi, *eLife*, 2018) where also they report the loss of granule cells. The main focus of this paper is the effect of agranularity on Purkinje cells and motor behaviors.

Using immunohistochemistry, authors show that Purkinje neurons are present, but have stunted dendritic arbors and lack clear organization into ZebrinII stripes as compared to controls. They also seem to receive more vGluT2 positive synapses (presumably from mossy fibers and climbing fibers), while lacking the vGluT1 positive parallel fiber synapses. Next authors record extracellularly from Purkinje cells of control mice during the first two post-natal weeks and from the agranular mice at P14. Purkinje neurons show a less bursty and irregular simple spike firing pattern in the agranular mice compared to control, which seems to be similar to the immature activity pattern seen at P8. Complex spikes are also less frequent. Finally authors show that the agranular mice exhibit several severe motor deficits including deficits in righting reflex, ultrasonic vocalizations and the presence of tremor.

These results, documenting severe impact on Purkinje neuron development, physiology and behavior are in contrast to earlier studies which showed mild effects on Purkinje neuron physiology after acute silencing of granule cell input. Thus the authors argue that the developmental presence of granule cells is more important in sculpting the Purkinje cell layer rather than just the excitatory synaptic input they provide.

While these ideas are compelling, I was left somewhat disappointed by this paper as there aren't any clear threads emerging from the various lines of investigation, nor are there links that connect them directly. What is the connection between data presented in figures 2,3 and 4? It is unclear which of these deficits reported in these figures are a direct result of not having granule cells and which are due to secondary effects arising out of compensatory and/or other pathways that may be affected. For instance, with granule cells being absent, feed forward inhibition in the molecular layer is also likely to be affected. With the behavioral readouts, effect of the mutations on extracerebellar structures cannot be ruled out either.

Another point is that it is also not clear whether any or all of these changes are merely a delay in development and whether they persist into adulthood. The authors should consider repeating some of their assays in the adults.

Figure 2 needs quantitative measures, especially with statements about the density of synapses. The argument that the increased vGlut2 synapses could be from retained mossy fibers is not investigated further, but needs to be.

For figure 3, there are a number of methods available to better estimate the regularity of neural activity, such as looking at power spectral density. Peak frequency and power at that frequency should be estimated for controls and mutants.

Are granule cells completely absent? The staining in Figure supplement 1A for the mutants appears weak and diffuse, such as would be seen if some granule cells were indeed present, but not organized into a layer. Authors should consider using another marker to show the agranularity as this is a major point of their paper. Likewise quantifications are necessary for making points such as "reduced number of UBCs". Are the numbers of Purkinje cells normal?

The discussion tries to place the study in the context of cerebellar pathologies in pre-term infants however, some of the statements are very strong and need to be toned down (such as that in lines 255-258). Authors should also try to address how they think the agranular state is influencing the properties of Purkinje neurons as described in the figures. In addition they should clearly state how the different results are related or not. Authors mention seeing lower number of complex spikes in mutants, but what they think this means is not discussed at all. Same goes for the doublets and for the zebrin stripes.

*Reviewer #2:*

This manuscript by van der Heijden et al. presents an interesting and thorough analysis of the cerebellar deficits in a novel mouse model, in which the transcription factor Atoh1 is conditionally deleted in Engrailed 1-expressing cells. Given what we know about cerebellar development and the consequences of disrupting this complex process from previous mouse models, the results of the study were largely to be expected. However, as the authors elegantly demonstrate, their new model is highly specific and reproducible and allows the interrogation of effects mostly specific to granule cell neurogenesis.

*Reviewer #3:*

In this article, van der Heijden et al. use mouse genetics to generate a model in which the transcription factor Atoh1 is deleted from the En1 domain and the cerebellum lacks granule cells. They analyze the consequences on cerebellar development, Purkinje cell firing properties and mouse behavior. They conclude that "granule cell neurogenesis sets the maturation time window for Purkinje cell function and refines cerebellar-dependent behavior." The mouse model generated by the authors provides a controlled way to generate agranular mutants and its analysis can provide new information on the mechanisms controlling the development of the cerebellar circuit. The morphological and behavior analysis confirms observations and conclusions made previously by several other groups through comparisons of different agranular models. The role of granule cells in the maturation of PC firing properties is an interesting new observation that deserves to be documented. However more work is needed to confirm that the phenotype is a blockade of maturation and to determine how granule cells control the development of this property.

1. The first conclusion of the manuscript is that lack of granule cells due to deletion of Atoh1 in the En1 domain leads to abnormal cerebellar development, Purkinje cell dendritic differentiation and circuit defects. While figure 1 clearly illustrates the major effect on cerebellar morphogenesis, the effects on the different neuronal populations deserves better documentation. The results of some of the immunostainings are not sufficient to document the lack of neurons (for example GABA-Ra6 or HCN1), and quantifications of the effects on cell numbers are not provided. In particular since unipolar brush cells are affected by the mutation, the effect should be quantified. The result on cerebellar development and Purkinje cell differentiation and connectivity is expected from previous research using x-irradiation and genetic mutants (these need to be cited in the Van der Heijden manuscript). Pan et al. (Cell Tissue Res (2009) 337:407-428) have shown that lack of granule cell death (induced by NeuroD1 knockout in the Atoh1 lineage) leads to defects in PC dendritogenesis and deficient mossy fiber targeting. Granule cell depletion using X-irradiation during early postnatal development in the rat has shown in addition defects in climbing fiber and mossy fiber connectivity both by electrophysiological and morphological analysis (cf. not only the cited reference Altman and Anderson 1971, but also for example Puro and Woodward 1978, Crepel and Delhaye-Bouchaud 1979, Arsenio Nunes et al. 1988).

2. The second conclusion of the manuscript is that lack of granule cells blocks the maturation of Purkinje cell firing. The result is interesting: the Purkinje cell firing properties are not similar to controls at two weeks and resemble those that are found in controls at the end of the first postnatal week. However a major issue is that measurements of PC firing were done only at P14 in the mutant and compared to several ages in controls (cf. figure 3). This leads to a problem in interpretation : is the phenotype really the result of blockade of maturation?

3. The third conclusion of the manuscript is that cerebellum dependent behaviors are deficient in these agranular mutants. Given the morphological and circuit phenotype, this is expected. The authors argue that "the contribution of the cerebellum to these behaviors is established before circuit rewiring is completed" to justify their analysis. However, what they mean by this is unclear and references to support this affirmation are not provided.

[Editors’ note: further revisions were suggested prior to acceptance, as described below.]

Thank you for resubmitting your work entitled "Maturation of Purkinje cell firing properties relies on neurogenesis of excitatory neurons" for further consideration by *eLife*. Your revised article has been reviewed by three peer reviewers, one of whom is a member of our Board of Reviewing Editors, and the evaluation has been overseen by K VijayRaghavan as the Senior Editor.

The manuscript has been improved but there are some remaining issues that need to be addressed, as outlined below:

The revised version of this manuscript includes new quantification and analysis and is a significant improvement. But, there are several points that remain to be clarified in the text or addressed with experiments and analysis. Please see the reviews below for a detailed description of these points.

*Reviewer #1:*

In this manuscript, authors generate mice lacking cerebellar granule cells using intersectional genetics and analyze the phenotype of these mice at the anatomical, physiological and behavioral levels. The major findings are that Purkinje neuron morphological and physiological development is impaired when granule cells are absent. In addition, motor coordination and ultrasonic vocalizations are also affected. These deficits are much more severe than the effects observed after acute silencing of granule cells in other studies. Thus, in addition to the synaptic influence of granule cells, their developmental presence at the correct stages seems to be critical for proper Purkinje cell development.

The manuscript is a significant improvement over the previous version that was submitted last year. Several results have now been properly quantified and add to the weight of their arguments. Nevertheless, some key points remain to be addressed:

1. The key result that the entire paper rests on is the agranular state of the En1Cre/+;Atoh1fl/- mice. The authors acknowledge that some granule cells and UBCs may escape and be present. The only data to show a lack of granule cells is the GABARa6 staining shown in Supp Figure 1-1A. As pointed out in the earlier review, given the criticality of knowing the extent of agranularity, authors should provide additional substantiation of this result. The ISH does not clarify whether the lower level of staining seen is non-specific background or remaining granule cells. This point needs to be clarified thoroughly. When the authors mention that there are escapees, how many cells do they see per animal?

2. Another point regarding genetic intervention is whether brain areas other than the cerebellum are affected. This is especially critical for understanding the basis of the behavioral phenotypes observed. Images showing ventricles and nuclear staining in key brain regions (cortex, hippocampus, striatum, etc) will help sort out this issue.

3. The authors claim that PCs in mutants are not bursty while those in controls at P14 are. The representative traces also show this clearly. Nevertheless, all of the analysis done is on spike regularity and do not reflect burstiness. The pause percent could reflect burstiness, but it will also depend on baseline firing rates, which are lower for the mutant. The autocorrelogram shows spike regularity within a +/- 100 ms window while the burst intervals are much longer and therefore are likely not reflected in the peaks of the autocorrelogram. The authors should consider Fourier analysis and show a heatmap power spectrum reflecting the various frequency bands found in PCs of P14 animals.

4. The authors' claims regarding Vglut1 and Vglut2 positive terminals onto PCs need more bolstering. Regarding Vglut1, the images shown in Figure 4 show diffuse weak staining of Vglut1 puncta. Can we rule out the existence of more widespread Vlglut1 synapses from escapee GCs and mossy fibers? In line 254, the authors state, "Taken together, we show that without excitatory cerebellar neurons, Purkinje cells do not acquire a large number of excitatory synapses during the second postnatal week." I do not believe that such a statement can be made based on these data on Vglut1 staining alone. To be able to say this, either mEPSC recordings or quantification of PF synapses using GFP positive lines must be done.

Similarly, for the Vglut2 staining, the authors state in line 368, "These data show that the rRL-derived excitatory cerebellar neurons are necessary for shaping the regional localization and cellular targeting of climbing fiber afferents that project from the inferior olive to the Purkinje cells." However, since Vglut2 is also present in mossy fibers, and it is not clear if the increased staining arises from an increased number of CFs projecting to single PCs, this is a strong interpretation of the results. For both vglut1 and vglut2 staining data, without additional experiments, effects on numbers or placement of CF/PF synapses cannot be unequivocally concluded.

*Reviewer #2:*

The authors have fully addressed my comments.

*Reviewer #3:*

This manuscript describes a smart genetic design to generate mice with agranular cerebella in a very reproducible manner and will be of interest to the field of cerebellar development. This mouse mutant recapitulates many previously described phenotypes for agranular mutants. A detailed in vivo analysis shows that Purkinje cells do not acquire mature firing properties, indicating a contribution of excitatory inputs to their development.

The manuscript from van der Heijden et al. describes the analysis of mice with an invalidation of the Atoh1 gene in the Engrailed 1 domain (mesencephalon and rhombomere 1) generated by breeding En1Cre mice with Atoh1Flox mice (En1^Cre/+^;Atoh1^fl/-^). This should result in the invalidation of Atoh1 in excitatory neurons generated by the rostral rhombic lip, and thus impair the generation of cerebellar granule cells. The authors use a broad range of techniques to provide a thorough analysis of the phenotype of these mice at the morphological, functional and behavioral levels.

Using immunolabeling with different cellular markers and quantification for some of the cell types, the authors demonstrate that indeed the majority of cerebellar granule cells are not generated and that the number of unipolar brush cells and excitatory cerebellar nuclei neurons is significantly decreased. The dendritogenesis of cerebellar PCs, as well as excitatory synapse formation on these neurons, are profoundly affected. These mutant mice display behavioral deficits: tremor, motor deficits and alterations of the pattern of ultrasonic vocalizations.

The main novelty is in the analysis of PC activity during postnatal development. The authors perform a detailed analysis during the 3 first postnatal weeks in control mice and analyze the properties of mutant PCs at P14 and P18. They show that during the second postnatal week a change occurs in the spontaneous activity of PCs in control mice. In En1^Cre/+^;Atoh1^fl/-^ mutants the properties of PCs at P14 and P18 more closely resemble those of Purkinje cells in control mice until P10. The authors thus propose that the switch in Purkinje cell activity is under the control of excitatory neurons in the cerebellum, and most likely granule cells.

1. The manuscript should clearly mention the fact that many of the phenotypes found in the En1^Cre/+^;Atoh1^fl/-^ mutants have already been described in other agranular models. While those models are less specific (genetic mutants) or less controlled (x-irradiation in neonates), their study has nonetheless led to similar findings. For example, in the X-irradiation model, the consequences have been described for PC dendritic development (cf. for review Sotelo and Dusart 2009); for circuit and synaptic maturation (multi-innervation by climbing fibers for example in Woodward, Hoffer and Altman, 1974 and Crepel, Delhaye-Bouchaud and Dupont 1981), ataxic behavior and tremor (Woodward, Hoffer and Altman, 1974) and Purkinje cell activity (also Woodward, Hoffer and Altman 1974). The similarities and differences between the phenotypes described in these papers and those in the van der Heijden manuscript should be discussed.

2. The mutant mice are called En1^Cre/+^/Atoh1^fl/-^ and the analysis is done by comparing to "control" mice in all the manuscript except in figure 9 where 3 different controls are used in the behavioral analysis. How were the mutants generated? Is there a cross with an Atoh1^-/-^(not mentioned in the methods)? What genotype is "control" mice in the manuscript?

3. The authors describe that there is a change in PC spiking during the second postnatal week in controls and that mutant PCs do not acquire the characteristics of control PCs. The study of the developmental pattern of PC spontaneous firing in vivo adds to previously obtained results by the team (Arancillo et al; 2015) and others (Crepel, 1972). The analysis done by van der Heijden et al. is detailed and measures many parameters. Their data confirm the observation of a switch during the second postnatal week (Crepel 1972 should be cited, in particular line 260). However, in figure 5, no statistical test is performed to determine which data points are significantly different between the different control stages analyzed. The use of clustering and t-SNE analysis is helpful and defines two broad stages: until P10 and after. But there seem to be a large heterogeneity amongst Purkinje cells at each postnatal day, and it is not exactly clear what drives this heterogeneity.

4. For the mutant PCs, the authors state that "this developmental switch does not occur by P14 in En1^Cre/+^;Atoh1^fl/-^mice". I do not think this can be stated since there is no data on the mutant PCs during the first postnatal week. It could be that the PCs in the agranular cerebellum never behave like control ones. The authors should record from PCs in the mutant at P7 if they want to state that the phenotype is due to the non-occurrence of a developmental switch.

5. There is sometimes confusion between intrinsic spiking properties and spontaneous spiking in Purkinje cells. For example, line 82 in Sathyanesan et al. 2018, the authors study spontaneous and evoked firing properties of PCs, not intrinsic properties. In the discussion, it is not very clear to me what the authors mean by "In addition to the increased complexity of intrinsic cellular properties (McKay and Turner, 2005), we postulated that intercellular interactions during development may support the maturation of Purkinje cell firing"(line 532/533). Could actually granule cells change the intrinsic spiking properties of PCs ? Slice recordings would be helpful to study this.

6. Line 358: There is a switch between Vglut2 and vglut1 at Parallel fiber PC synapses during development (cf. Miyazaki EJN 2003). The decrease in VGLUT2 puncta density observed in the molecular layer is most likely due to this switch rather that CF synaptic pruning. It is thus difficult to interpret the result in panel 7C.

[Editors’ note: further revisions were suggested prior to acceptance, as described below.]

Thank you for resubmitting your work entitled "Maturation of Purkinje cell firing properties relies on neurogenesis of excitatory neurons" for further consideration by *eLife*. Your revised article has been reviewed by 1 peer reviewer and the evaluation has been overseen by K VijayRaghavan as the Senior Editor, and a Reviewing Editor.

In this revised version, the authors have addressed most of the points raised by the reviewers. However, a few points remain and these are detailed below.

*Reviewer #3:*

The authors have answered many of my concerns and the manuscript is significantly improved. There remain however several points that need to be addressed before publication.

1. If I understand well the explanation about the control mice, the controls are either Atoh1^fl/+^, En1^Cre/+^;Atoh1^fl/+^ or Atoh1^fl/FlpO^ and the data are grouped together except in the behavioral analysis. This should be stated clearly not only in the methods section, but at the beginning of the Results section (line 138 and line 146), as it is important for the interpretation of the results (some of the variability in the "control" data could be due to the difference in genotype). As it is, the description is misleading. How was the Atoh1FlpO line characterized (I can only find genotyping data in a previous paper)? Is there any data showing that the three control genotypes are not different from each other and from wild-type animals?

2. Result section page 10 lines 342 to 345: Crepel had described in 1972 (Experimental Brain Research) in vivo recordings of Purkinje cell activity in rat neonates and showed increased frequency with maturation (see figure 1 of that paper). The Crepel paper should be cited in this part of the result section, as well as in the discussion page 19. Line 343, the Beekhof et al. article describes not only in vitro but also in vivo recordings of Purkinje cell activity. The sentence as it is in line 343 is misleading.

3. The last paragraph of the discussion should be simplified. Granule cells influence the maturation of Purkinje cell spiking (line 728-729), but also many other parameters of Purkinje cell development and maturation, as described in this manuscript but also previous data on agranular models. As the relationship between the different deficits is difficult to disentangle, this part of the discussion should be toned down (line 728 for example "direct influence").

---

## [Author Response]

[Editors’ note: the authors resubmitted a revised version of the paper for consideration. What follows is the authors’ response to the first round of review.]

Reviewer #1:[…] I was left somewhat disappointed by this paper as there aren't any clear threads emerging from the various lines of investigation, nor are there links that connect them directly. What is the connection between data presented in figures 2,3 and 4?

In our revised manuscript, we have clarified the connections between the cellular anatomy and electrophysiology data as well as network-level behavioral abnormalities by focusing on the changes that occur between P7 and P14. In control animals, there are highly dynamic changes in cerebellar size (figure 1), density in Vglut1 and Vglut2 synapses (figure 4 and 7), and, as reported before, electrophysiological properties in Purkinje cells (figure 5, 6, and 8). None of these changes occur in our agranular mice.

It is unclear which of these deficits reported in these figures are a direct result of not having granule cells and which are due to secondary effects arising out of compensatory and/or other pathways that may be affected. For instance, with granule cells being absent, feed forward inhibition in the molecular layer is also likely to be affected. With the behavioral readouts, effect of the mutations on extracerebellar structures cannot be ruled out either.

We appreciate that the lack of granule cells has additional, secondary effects. We have included a paragraph in our discussion outlining the caveats of our model (Line 451-474) and the considerations that readers should take when interpreting the behavioral data (Line 499-547). Thus, we present the strengths and new information that our model has provided, but we now also provide a more balanced assessment of the data and interpretations.

Another point is that it is also not clear whether any or all of these changes are merely a delay in development and whether they persist into adulthood. The authors should consider repeating some of their assays in the adults.

This is a great question. Unfortunately, our mouse model does not survive post weaning age, likely due to a combination of breathing impairments and severe motor impairments (see Van der Heijden and Zoghbi, 2018). However, we have addressed this issue by performing additional experiments. In lieu of performing additional recordings in adult mice, therefore, we have provided additional analysis to show that in control animals there is a striking functional switch in Purkinje cell function that does not occur in the same ages of our mutant mice (Figure 6). The temporal coincidence of this switch, with rearrangement of synapses at the peak of granule cell integration, further suggests that this maturation is driven by granule cells entering the circuit, which never occurs in the agranular mice. Additionally, we recorded from P18 conditional knockout and control mice. By this time point, the functional switch has still not occurred. These data are included in Figure 6 – supplement 1. Again, we could not reliably record from mice older than P18 as the health of the mutants rapidly declines towards the time of death, which happens consistently within one to three days.

Figure 2 needs quantitative measures, especially with statements about the density of synapses. The argument that the increased vGlut2 synapses could be from retained mossy fibers is not investigated further, but needs to be.

Thank you, this is a good idea. We have now quantified the density of Vglut1 and Vlgut2 synapses and have adjusted our text accordingly.

For figure 3, there are a number of methods available to better estimate the regularity of neural activity, such as looking at power spectral density. Peak frequency and power at that frequency should be estimated for controls and mutants.

We already include the peak frequency as the frequency mode in our figure. We decided to include a second measure of regularity by calculating the rhythmicity index, which is computed by the oscillation properties of the autocorrelation simple spikes in our Purkinje cells. High oscillatory properties in the autocorrelation means highly rhythmic/regular firing patterns, thus a high rhythmicity index is means high regularity. This computation is included in Figure 5I-K.

Are granule cells completely absent? The staining in Figure supplement 1A for the mutants appears weak and diffuse, such as would be seen if some granule cells were indeed present, but not organized into a layer. Authors should consider using another marker to show the agranularity as this is a major point of their paper. Likewise quantifications are necessary for making points such as "reduced number of UBCs". Are the numbers of Purkinje cells normal?

We quantified the number of UBCs and excitatory nuclei cells and included this quantification in Supplemental Figure 1-2. It is likely that the number of Purkinje cells is reduced, as we have discussed in Lines 469-474. However, our investigations are based on density of excitatory synapses and single cell recordings, neither of which are likely affected by the total number of Purkinje cells.

Finally, it remains a possibility that a small number granule cells escape our genetic manipulation. However, this number is likely very low because (1) the granule cells are the major cause for cerebellar size expansion, which we do not see in our *En1-Cre;Atoh1-fl/-* mice; (2) we see no systematic expression of the granule cell marker, GABARα6, in our *En1-Cre;Atoh1-fl/-* mice; (3) we see a nearly 9-fold decrease in Vglut1 density, which is overwhelmingly due to the presence of mossy fiber projections, as the remaining granule cells provide only a minor contribution. We have included these points in our discussion Lines 451-461.

The discussion tries to place the study in the context of cerebellar pathologies in pre-term infants however, some of the statements are very strong and need to be toned down (such as that in lines 255-258). Authors should also try to address how they think the agranular state is influencing the properties of Purkinje neurons as described in the figures. In addition they should clearly state how the different results are related or not. Authors mention seeing lower number of complex spikes in mutants, but what they think this means is not discussed at all. Same goes for the doublets and for the zebrin stripes.

We have added in the text more explanations about the relationship between agranularity, changes in excitatory synapses onto Purkinje cells, and observed electrophysiological changes. We have vastly expanded our analysis of Vglut1/2 densities onto Purkinje cells, that show that lower Vglut1 density is paired with lower simple spike firing rate, whereas higher and more immature-like Vglut2 density is paired with lower complex spike number and higher number of doublets. We have also revised and expanded our discussion about the ZebrinII zones.

Reviewer #3:[…] 1. The first conclusion of the manuscript is that lack of granule cells due to deletion of Atoh1 in the En1 domain leads to abnormal cerebellar development, Purkinje cell dendritic differentiation and circuit defects. While figure 1 clearly illustrates the major effect on cerebellar morphogenesis, the effects on the different neuronal populations deserves better documentation. The results of some of the immunostainings are not sufficient to document the lack of neurons (for example GABA-Ra6 or HCN1), and quantifications of the effects on cell numbers are not provided. In particular since unipolar brush cells are affected by the mutation, the effect should be quantified. The result on cerebellar development and Purkinje cell differentiation and connectivity is expected from previous research using x-irradiation and genetic mutants (these need to be cited in the Van der Heijden manuscript). Pan et al. (Cell Tissue Res (2009) 337:407-428) have shown that lack of granule cell death (induced by NeuroD1 knockout in the Atoh1 lineage) leads to defects in PC dendritogenesis and deficient mossy fiber targeting. Granule cell depletion using X-irradiation during early postnatal development in the rat has shown in addition defects in climbing fiber and mossy fiber connectivity both by electrophysiological and morphological analysis (cf. not only the cited reference Altman and Anderson 1971, but also for example Puro and Woodward 1978, Crepel and Delhaye-Bouchaud 1979, Arsenio Nunes et al. 1988).

We now quantified the number of unipolar brush cell, excitatory nuclei cells, and Vglut1/2 density. Please also see our responses above to Reviewer #1 and Reviewer #2.

We appreciate that this reviewer recognizes that our mouse model recapitulates many aspects of previously reported agranular mice. This was the intention of some of our experiments, as we needed to validate our *new* model for cerebellar agranularity. We specifically choose this model of agranularity because it was reliable, independent of experimental variability, and independent of genes expressed in Purkinje cells. The last was specifically important because we wanted to exclude the possibility that our results were confounded by Purkinje cell autonomous effects. The fact that many of the anatomical findings are similar to those reported in other models of agranularity, underscores the effectiveness of our new genetic approach. We have now included more references to previous models of agranularity to highlight the similarities to our new models. Specifically, we have added the excellent references the reviewer has suggested – thank you.

2. The second conclusion of the manuscript is that lack of granule cells blocks the maturation of Purkinje cell firing. The result is interesting: the Purkinje cell firing properties are not similar to controls at two weeks and resemble those that are found in controls at the end of the first postnatal week. However a major issue is that measurements of PC firing were done only at P14 in the mutant and compared to several ages in controls (cf. figure 3). This leads to a problem in interpretation : is the phenotype really the result of blockade of maturation?

This is an excellent point, and one that was also raised by Reviewer #1 above. We noted above that unfortunately, our mouse model does not survive post weaning age, likely due to a combination of breathing impairments and severe motor impairments (see Van der Heijden and Zoghbi, 2018). However, we have addressed this issue by performing additional experiments. In lieu of performing additional recordings in adult mice, therefore, we have provided additional analysis to show that in control animals there is a striking functional switch in Purkinje cell function that does not occur in the same ages of our mutant mice (Figure 6). The temporal coincidence of this switch, with rearrangement of synapses at the peak of granule cell integration, further suggests that this maturation is driven by granule cells entering the circuit, which never occurs in the agranular mice. Additionally, we recorded from P18 conditional knockout and control mice. By this time point, the functional switch has still not occurred. These data are included in Figure 6 – supplement 1. Again, we could not reliably record from mice older than P18 as the health of the mutants rapidly declines towards the time of death, which happens consistently within one to three days.

3. The third conclusion of the manuscript is that cerebellum dependent behaviors are deficient in these agranular mutants. Given the morphological and circuit phenotype, this is expected. The authors argue that "the contribution of the cerebellum to these behaviors is established before circuit rewiring is completed" to justify their analysis. However, what they mean by this is unclear and references to support this affirmation are not provided.

We included a more in-depth discussion of our findings in the behavioral essays in our discussion:

“The behavioral abnormalities in *En1^Cre/+^;Atoh1^fl/-^* mice are likely caused by a combined effect of Purkinje cell dysfunction and a reduction of excitatory neurons in the cerebellar nuclei. Given the broad network effects of our manipulation, it is not surprising that *En1^Cre/+^;Atoh1^fl/-^* mice have severe behavioral abnormalities. […] This shows that unlike Purkinje cells, excitatory cerebellar neurons are not necessary for propagating tremor but are required for modulating it.”

[Editors’ note: what follows is the authors’ response to the second round of review.]

Reviewer #1:[…] The manuscript is a significant improvement over the previous version that was submitted last year. Several results have now been properly quantified and add to the weight of their arguments. Nevertheless, some key points remain to be addressed:1. The key result that the entire paper rests on is the agranular state of the En1^Cre/+^;Atoh1^fl/-^ mice. The authors acknowledge that some granule cells and UBCs may escape and be present. The only data to show a lack of granule cells is the GABARa6 staining shown in Supp Figure 1-1A. As pointed out in the earlier review, given the criticality of knowing the extent of agranularity, authors should provide additional substantiation of this result. The ISH does not clarify whether the lower level of staining seen is non-specific background or remaining granule cells. This point needs to be clarified thoroughly. When the authors mention that there are escapees, how many cells do they see per animal?

To address this comment, we decided to employ a genetic strategy to circumvent the potential ISH issues, as pointed out by the reviewer. In our genetic strategy, we marked all *En1;Atoh1* lineage neurons with TdTomato using an intersectional allele (see Supp. Figure 1-3). In this method, the marker signal of granule cells in the granule cell layer of control animals is near saturation (which was fully expected), due the density of their cell bodies (99.6% ± 0.4% of the area of interest measured is occupied by the TdTomato signal). However, we were able to distinguish individual TdTomato^+^ neurons in the conditional knockout mice and estimated that ~65 thousand neurons escaped our genetic manipulation. This is an approximate thousand-fold reduction in neurons compared to what has previously been reported for control animals in the literature (between 35 and 70 million neurons using density estimations or isotropic fractionation (Vogel et al. 1989; Herculano-Houzel et al. 2006; Surchev et al. 2007; Consalez et al. 2020)).

We now provide several lines of evidence that all indicate and describe the significant reduction of granule cells in our conditional knockout mice:

1) The total number of *Atoh1;En1* lineage neurons that are left in the mutant is in the range of tens of thousands of cells (versus tens of millions reported in the literature for control animals).

2) There is a nearly 50-fold reduction in cerebellar size (10X in sagittal plane, 5X in coronal plane), and the cerebellum does not have a visible granule cell layer or obvious cerebellar foliation. These morphological phenotypes encompass the most experimentally consistent and profound cerebellar malformations as a result of granule cell agenesis, and together culminate into a severe phenotype compared to previously reported manipulations. (X-irradiated rats: small cerebellar size, thin GCL (Zanjani et al. 1992; Armstrong and Hawkes 2001); *staggerer* and *weaver* mice: small cerebellar size, thin GCL, less foliation (Zanjani et al. 1992; Armstrong and Hawkes 2001); *reeler* and *scrambler* mice: small cerebellar size, thin GCL, no foliation (Goldowitz et al. 1997); *Atoh1* chimera mice: small cerebellar size, thin/no GCL, less/no foliation (phenotype is variable depending on degree of chimerism) (Jensen et al. 2004)). Our mutants exhibit each of these phenotypes, but with increased severity.

3) The total density of Vglut1 signal is reduced 9-fold in the mutant mice (Figure 4). However, this number does not only account for the total escaper granule cells because the density analysis does not distinguish the granule cell parallel fiber synapses from either Vglut1 expression in mossy fibers or escaper UBCs, and in addition this estimate does not account for the overall size difference of the cerebellum.

4) Purkinje cell monolayer formation and the establishment of mature stripe/zonal patterns do not occur (Figure 1 and 2).

5) VGluT2^+^ climbing fibers and/or mossy fiber synapse displacement from the Purkinje cell bodies is severely impaired (Figure 2 and 7).

Based on these different but complementary lines of evidence, we conclude that our genetic model of agranularity significantly reduces granule cell neurogenesis to ultimately result in the cerebellum having only a fraction of differentiated granule cells. The manipulation therefore massively obstructs but does not completely eliminate granule cell neurogenesis.

2. Another point regarding genetic intervention is whether brain areas other than the cerebellum are affected. This is especially critical for understanding the basis of the behavioral phenotypes observed. Images showing ventricles and nuclear staining in key brain regions (cortex, hippocampus, striatum, etc) will help sort out this issue.

This is an excellent suggestion. We have generated a new supplemental Figure (Supp Figure 1-1 A-B) showing that the overt gross morphological changes only occur in the cerebellum.

3. The authors claim that PCs in mutants are not bursty while those in controls at P14 are. The representative traces also show this clearly. Nevertheless, all of the analysis done is on spike regularity and do not reflect burstiness. The pause percent could reflect burstiness, but it will also depend on baseline firing rates, which are lower for the mutant. The autocorrelogram shows spike regularity within a +/- 100 ms window while the burst intervals are much longer and therefore are likely not reflected in the peaks of the autocorrelogram. The authors should consider Fourier analysis and show a heatmap power spectrum reflecting the various frequency bands found in PCs of P14 animals.

Apologies for creating confusion in our descriptions of the “burstiness” in the parameters quantified in Figure 5. We refer to “bursty” cells as cells that have many high-frequency (short inter-spike interval (ISI)) spikes occurring in bursts that are interspersed by a few low frequency (long ISI) pauses. The vast majority of the spike trains thus have a short ISI, whereas a small proportion in our cells have a very long ISI. The reviewer is correct that these bursts are not reflected in the regularity within a +/- 1 s window (as is reflected in Figure 5L), because the bursts occur at a longer interval, have variable length, and only a small proportion of all the spikes have a very long ISI, despite their clear contribution to the burst. Therefore, we would not see a peak for the long ISI spikes in the auto correlogram even if we extended the time window beyond 1 s.

Our parameters define two types of irregularities: global and local. Global irregularity is caused by the occurrence of pauses and bursts (as in the Purkinje cells in P14 control mice). Local irregularity is caused by irregularity of inter spike intervals (as in the Purkinje cells of P14 mutant mice).

To examine the burstiness (or global irregularity in firing patterns) in our Purkinje cells we used a number of other parameters that are very effective for detecting global changes in neuronal firing patterns:

1. Frequency is calculated as the total number of spikes per second. Frequency mode (or preferred firing frequency) is calculated as the inverse of the most common ISI. We are able to detect the presence of “bursty” features because the presence of very long ISIs alters the relationship between Frequency and Frequency mode since the long ISIs reduce the average Frequency that is calculated over the entire duration of the recording. For example, the Purkinje cells in P13-14 control animals have an average Frequency of ~13.7 Hz, whereas the Frequency mode is ~26.2 Hz (nearly 2-fold difference). In contrast, in the P14 conditional knockout mice the Purkinje cell Frequency is ~4.35 Hz and the Frequency mode is ~5.25 Hz (about 1.2-fold difference). Therefore, Frequency-based relationships can reflect the pattern.

2. As the reviewer states, the pause proportion reflects burstiness. We defined a pause as an ISI that was five times as long as the most common ISI, to control for the differences in firing rate that the reviewer has mentioned. This calculation allowed us to measure the more reliable breaks in between spikes instead of long ISI that can happen in slowly firing neurons. If we would define the pause as an ISI of pre-set duration, it would miss pauses in fast-firing neurons or designate regularly occurring long ISI in slow firing neurons a pause. For example, in Purkinje cells in the P13-14 control animals have a pause proportion of ~0.35 whereas Purkinje cells in P14 conditional knockout mice have a pause proportion of ~0.14.

3. Finally, we included the CV. This parameter is calculated as the ratio of the standard deviation and mean of the ISI. As mentioned before, in the bursty cells, there are a few ISI that are much longer than the most common ISI (namely the pauses). Thus, the range, and therefore the standard deviation, of ISIs in bursty cells is very large. By dividing this large standard deviation by the mean ISI we normalize this variability to the firing rate to give a proportional variation. As a result, a more bursty cell will have a higher CV. For example, in Purkinje cells in the P13-14 control animals the CV is ~4.0 whereas in P14 conditional knockout mice, Purkinje cells in have a CV of ~1.3.

Even though we find that Purkinje cells in the P13-14 control animals have more bursty firing patterns, we observe that the simple spikes *within* the burst are very rhythmic. This is the reason we also included measures of local spike irregularity namely, the rhythmicity index (low in irregular cells) and CV2 (high in irregular cells). We now define in the text that these are measures for local irregularity in firing patterns.

We liked the idea that this reviewer has raised. The suggestion was to show a heatmap of the various frequencies at which the cells fire. Unfortunately, performing a Fourier analysis on single cell in vivo recordings would not provide us with an easily interpretable heatmap. Even though our single cell recordings are continuous, the signal information in our continuous recording is binary; spike or no spike. As a result, the Fourier analysis would detect the sinusoidal shape of single spikes, but not the frequency at which these spikes occur. Performing a Fourier analysis on the frequency averaged over small time-windows (for example 1 s as is presented in Figure 5H and I) would not provide a satisfactory power frequency plot because the pauses and bursts occur at random intervals. Instead, to visualize a heatmap of various frequency bands, as the reviewer suggested, we included heatmaps of spike frequency distributions of all 149 cells included in Figure 5 and 6 as Figure 5 – supplement 1. In this figure, some cells show a second, small peak, at a lower frequency, which indicates pause-events (and therefore bursts). However, the firing frequency is much higher within the burst than within the pause (per definition) and the duration of the pause is not the same for each pause. Therefore, the relative proportion of spikes that occur at a low frequency is so small that this second peak in not visible for all cells.

4. The authors' claims regarding Vglut1 and Vglut2 positive terminals onto PCs need more bolstering. Regarding Vglut1, the images shown in Figure 4 show diffuse weak staining of Vglut1 puncta. Can we rule out the existence of more widespread Vlglut1 synapses from escapee GCs and mossy fibers? In line 254, the authors state, "Taken together, we show that without excitatory cerebellar neurons, Purkinje cells do not acquire a large number of excitatory synapses during the second postnatal week." I do not believe that such a statement can be made based on these data on Vglut1 staining alone. To be able to say this, either mEPSC recordings or quantification of PF synapses using GFP positive lines must be done.

We agree with the reviewer. It is an excellent suggestion to bolster our statement that the number of excitatory synapses on Purkinje cells is reduced. Therefore, to address this comment, we quantified the spine density on Golgi-Cox stained images of Purkinje cells from control and mutant cerebellar. These quantifications are now included in Figure 5D-E.

Similarly, for the Vglut2 staining, the authors state in line 368, "These data show that the rRL-derived excitatory cerebellar neurons are necessary for shaping the regional localization and cellular targeting of climbing fiber afferents that project from the inferior olive to the Purkinje cells." However, since Vglut2 is also present in mossy fibers, and it is not clear if the increased staining arises from an increased number of CFs projecting to single PCs, this is a strong interpretation of the results. For both vglut1 and vglut2 staining data, without additional experiments, effects on numbers or placement of CF/PF synapses cannot be unequivocally concluded.

This is a good point. We reworded the cited sentence to include mossy fibers. The new sentence now reads:

“These data show that the *En1;Atoh1* lineage, excitatory cerebellar neurons are necessary for shaping the regional localization and cellular targeting of immature mossy fibers and climbing fiber afferents that project to the Purkinje cells.”

Reviewer #3:[…] 1. The manuscript should clearly mention the fact that many of the phenotypes found in the En1^Cre/+^;Atoh1^fl/-^ mutants have already been described in other agranular models. While those models are less specific (genetic mutants) or less controlled (x-irradiation in neonates), their study has nonetheless led to similar findings. For example, in the X-irradiation model, the consequences have been described for PC dendritic development (cf. for review Sotelo and Dusart 2009); for circuit and synaptic maturation (multi-innervation by climbing fibers for example in Woodward, Hoffer and Altman, 1974 and Crepel, Delhaye-Bouchaud and Dupont 1981), ataxic behavior and tremor (Woodward, Hoffer and Altman, 1974) and Purkinje cell activity (also Woodward, Hoffer and Altman 1974). The similarities and differences between the phenotypes described in these papers and those in the van der Heijden manuscript should be discussed.

At the beginning of our Results section, we have now included a paragraph describing the gross anatomical malformations in agranular animal models, as this is the most consistently reported phenotype described across all such models.

We have also included additional citations throughout the text to indicate the similarities and differences between the cellular-level phenotypes we observe in our mice and those reported in previous publications on models of cerebellar agranularity.

2. The mutant mice are called En1^Cre/+^/Atoh1^fl/-^ and the analysis is done by comparing to "control" mice in all the manuscript except in figure 9 where 3 different controls are used in the behavioral analysis. How were the mutants generated? Is there a cross with an Atoh1^-/-^(not mentioned in the methods)? What genotype is "control" mice in the manuscript?

We did not do the null cross because *Atoh1^-/-^* mice are neonatal lethal. We have now clarified this in the methods as follows:

“Conditional knockout mice were generated by crossing *En1^Cre/+^;Atoh1^+/FlpO^* double heterozygote mice (which can be functionally defined as *En1^Cre/+^;Atoh1^+/-^* mice) with homozygote *Atoh1^fl/fl^* mice. *En1^Cre/+^;Atoh1^fl/-^* mice were considered experimental conditional knockout mice and all littermates with a different genotype were considered control mice.”

3. The authors describe that there is a change in PC spiking during the second postnatal week in controls and that mutant PCs do not acquire the characteristics of control PCs. The study of the developmental pattern of PC spontaneous firing in vivo adds to previously obtained results by the team (Arancillo et al; 2015) and others (Crepel, 1972). The analysis done by van der Heijden et al. is detailed and measures many parameters. Their data confirm the observation of a switch during the second postnatal week (Crepel 1972 should be cited, in particular line 260). However, in figure 5, no statistical test is performed to determine which data points are significantly different between the different control stages analyzed. The use of clustering and t-SNE analysis is helpful and defines two broad stages: until P10 and after. But there seem to be a large heterogeneity amongst Purkinje cells at each postnatal day, and it is not exactly clear what drives this heterogeneity.

This is an excellent suggestion. To attain the necessary power to make the statistical comparisons between all the age groups, we grouped together our data from P7-8, P9-10, P11-12, and the P13-14 mice. We next performed an ANOVA between all groups (4 control groups, 2 experimental groups) and did a post-hoc analysis to identify which groups have a statistically significant different from each other.

The use of clustering and t-SNE analysis is helpful and defines two broad stages: until P10 and after. But there seem to be a large heterogeneity amongst Purkinje cells at each postnatal day, and it is not exactly clear what drives this heterogeneity.

Heterogeneity in the maturation of anatomical properties have been reported previously (McKay and Turner, 2005; van Welie et al., 2011). Furthermore, it is known that adult Purkinje cells that are recorded in vivo in awake animals have a large degree of variability in their firing frequency and regularity (see for example our previous publications: Brown et al., 2020; Miterko et al., 2021; van der Heijden et al., 2021). Other groups have found that this heterogeneity can be in part attributed to the molecular identity (patterning) of Purkinje cells (Zebrin^+^ vs. Zebrin^-^ cells) (Zhou et al., 2014; Wu et al., 2019). It was also recently reported that the patterned identity of Purkinje cells within specific zones arises according to the development of these patterns and ultimately contributes to the heterogeneity of Purkinje cell functional properties observed in adults (Beekhof et al., 2021). Taken together, the heterogeneity in Purkinje cell function is an expected observation. We address this point in the text as follows:

“This suggests the existence of a transformation that occurs between younger and older cells, but the transformation does not occur at the same time for all Purkinje cells, as is expected based on previous studies showing that the heterogeneity of Purkinje cell function is an inherent property of Purkinje cells that respects and spans the timelines of anatomical and functional development of Purkinje cells (Beekhof et al. 2021; van Welie et al. 2011; McKay and Turner 2005).”

Additionally, we account for the expected inter- and intra-animal heterogeneity of Purkinje cell development by recording from a large number of single neurons (10-20) and from multiple mice (3-7) on each of the 8 developmental days that fall within our analysis time-window (P7 to P14).

4. For the mutant PCs, the authors state that "this developmental switch does not occur by P14 in En1^Cre/+^;Atoh1^fl/-^mice". I do not think this can be stated since there is no data on the mutant PCs during the first postnatal week. It could be that the PCs in the agranular cerebellum never behave like control ones. The authors should record from PCs in the mutant at P7 if they want to state that the phenotype is due to the non-occurrence of a developmental switch.

This is another excellent suggestion. We decided to perform additional recordings in P10 mutant mice. We choose this time point for several reasons:

1: Our mutant mice lack several excitatory pontine respiratory nuclei (Van der Heijden and Zoghbi, 2018), which makes them more sensitive to the respiratory repression that results from anesthetics such as ketamine, which we use in our studies for anesthetizing the mice during our recordings. In our experience, P10 mutant mice were more resilient to the anesthesia than P7 mutant mice. Using the P10 timepoint allowed us to record mice using the same dose as all other mice in the study, which ultimately helped us avoid having to change the anesthesia concentration, which would have been a confounding factor in our analysis.

2: The initial ANOVA results as described in the first part of this reviewer comment showed no statistically significant difference between P7 and P10 control mice.

3: Our clustering analysis showed that P14 experimental mice were most similar to P10 control mice (Figure 6D).

When we performed these experiments, we found that Purkinje cells in P10 mutant mice were not different from Purkinje cells in P14 mutant mice in any of the parameters included in our study, confirming our hypothesis that the developmental transformation that typically occurs at this stage does not occur in our mutant mice.

5. There is sometimes confusion between intrinsic spiking properties and spontaneous spiking in Purkinje cells. For example, line 82 in Sathyanesan et al. 2018, the authors study spontaneous and evoked firing properties of PCs, not intrinsic properties. In the discussion, it is not very clear to me what the authors mean by "In addition to the increased complexity of intrinsic cellular properties (McKay and Turner, 2005), we postulated that intercellular interactions during development may support the maturation of Purkinje cell firing"(line 532/533). Could actually granule cells change the intrinsic spiking properties of PCs ? Slice recordings would be helpful to study this.

The reviewer is correct. We did not mean to draw too many conclusions with reference to intrinsic properties as our in vivo recordings did not measure such firing features. To avoid this confusion, we changed the wording from intrinsic to spontaneous. We have also toned down the extent of comparisons that we make to the previous slice physiology data from the Turner lab. We have now made the intent of our comparison clearer:

“In addition to the increased complexity of intrinsic cellular properties that have been defined using slice electrophysiology approaches (McKay and Turner 2005), we postulated using our in vivo recordings that intercellular interactions between granule cells and Purkinje cells during development may support the maturation of Purkinje cell firing properties, both by providing direct inputs and by supporting anatomical maturation, including the outgrowth of Purkinje cell dendrites.”

6. Line 358: There is a switch between Vglut2 and vglut1 at Parallel fiber PC synapses during development (cf. Miyazaki EJN 2003). The decrease in VGLUT2 puncta density observed in the molecular layer is most likely due to this switch rather that CF synaptic pruning. It is thus difficult to interpret the result in panel 7C.

This is a good point. We included this citation in the text. We clarified the interpretation of figure panel 7C as followed:

“Immature VGluT2^+^/climbing fiber synapses are located around the Purkinje cell body in controls, but at around P9 only the “winner” among these synapses translocates to the Purkinje cell dendrites in the molecular layer (Kano et al. 2018). […] These data show that the *En1;Atoh1* lineage, excitatory cerebellar neurons are necessary for shaping the regional localization and cellular targeting of immature mossy fibers and climbing fiber afferents that project to the Purkinje cells.”

[Editors' note: further revisions were suggested prior to acceptance, as described below.]

In this revised version, the authors have addressed most of the points raised by the reviewers. However, a few points remain and these are detailed below.Reviewer #3:The authors have answered many of my concerns and the manuscript is significantly improved. There remain however several points that need to be addressed before publication.1. If I understand well the explanation about the control mice, the controls are either Atoh1^fl/+^, En1Cre/+;Atoh1^fl/+^ or Atoh1^fl/FlpO^ and the data are grouped together except in the behavioral analysis. This should be stated clearly not only in the methods section, but at the beginning of the Results section (line 138 and line 146), as it is important for the interpretation of the results (some of the variability in the "control" data could be due to the difference in genotype). As it is, the description is misleading.

Thank you for pointing out this confusion, we addressed this concern as followed in the start of the result section:

“We generated conditional knockout mice by crossing mice that are heterozygous for the *En1^Cre^* knock-in allele (Wurst et al., 1994) and heterozygous for *Atoh1* (van der Heijden and Zoghbi, 2018) with mice that are homozygous for a LoxP-flanked *Atoh1* allele (*Atoh1^fl/fl^*) (Shroyer et al., 2007). From this cross we obtain mice with four genotypes: *Atoh1^fl/+^* mice, *Atoh1^FlpO/fl^* mice, and *En1^Cre/+^;Atoh1^fl/+^* mice that we hereafter will refer to as control mice, and our experimental *En1^Cre/+^;Atoh1^FlpO/+^
*mice that we hereafter will refer to as *En1^Cre/+^;Atoh1^fl/-^* mice because the *Atoh1^FlpO^* is a functional null-allele (van der Heijden and Zoghbi, 2018).”

How was the Atoh1FlpO line characterized (I can only find genotyping data in a previous paper)?

The *Atoh1^FlpO^* line is characterized and validated by (1) PCR validation of correct insertion in the endogenous *Atoh1*-locus and loss of the coding region for the *Atoh1* gene; (2) expression of the *Atoh1^FlpO^*-allele in the brainstem in this and our previous paper (Van der Heijden and Zoghbi, 2018); (3) extensive outcrossing of the founder mice to avoid potential other genomic rearrangements confounding our expression; (4) validation that the *Atoh1^FlpO/+^* mice have no unexpected behavioral phenotypes included those discussed in Figure 9 of the current paper and extensive breathing assays in our previous paper.

Is there any data showing that the three control genotypes are not different from each other and from wild-type animals?

We would expect that if there were any biologically meaningful differences between either of the control groups it would first show up at the mouse behavioral level as the *Atoh1^FlpO/+^* mice are heterozygote for *Atoh1* in all cells, and *En1^Cre/+^;Atoh1^fl/+^* are heterozygote for *En1* in all cells and *Atoh1* only in the *En1-*lineage. Nevertheless, we find no significant differences in any of the behavioral assays described in Figure 9 (see also the raw P-values in extended data for Figure 9).

2. Result section page 10 lines 342 to 345: Crepel had described in 1972 (Experimental Brain Research) in vivo recordings of Purkinje cell activity in rat neonates and showed increased frequency with maturation (see figure 1 of that paper). The Crepel paper should be cited in this part of the result section, as well as in the discussion page 19. Line 343, the Beekhof et al. article describes not only in vitro but also in vivo recordings of Purkinje cell activity. The sentence as it is in line 343 is misleading.

Thank you for pointing out that we should include the Crepel 1972 in our references. As for the Beekhof et al., 2021 citation, all in vivo recordings were performed in mice with earliest timepoint P12 *through* P17, but not earlier. To clarify, and avoid any potential confusion for the reading, we included the timepoints relevant to our paper in this sentence. The sentence now reads:

“in vitro recordings in rats (Crepel, 1972; McKay and Turner, 2005) and mice (Beekhof et al., 2021) show that Purkinje cell firing properties change significantly during early postnatal development (P7-P14), but it is unclear how firing patterns evolve in vivo during this dynamic period of rewiring.”

3. The last paragraph of the discussion should be simplified. Granule cells influence the maturation of Purkinje cell spiking (line 728-729), but also many other parameters of Purkinje cell development and maturation, as described in this manuscript but also previous data on agranular models. As the relationship between the different deficits is difficult to disentangle, this part of the discussion should be toned down (line 728 for example "direct influence").

We reworded this sentence as followed:

“Based on our data, we argue that the initial communication between Purkinje cells and granule cells sets the efficiency of Purkinje cell function and the establishment of Purkinje cell spike properties through structural as well as synaptic signals.”